# Sweet and sticky: increased cell adhesion through click-mediated functionalization of regenerative liver progenitor cells
Amaziah R. Alipio [1], Melissa R. Vieira [2,3], Tamara Haefeli[4], Lisa Hoelting[4], Olivier Frey[4], Alicia J. El Haj [2,3] ✉ & Maria C. Arno [1,5] ✉

The burgeoning field of cell therapies is rapidly expanding, offering the promise to tackle complex and unsolved healthcare problems. One prominent example is represented by CAR T-cells, which have been introduced into the clinic for treating a variety of cancers. Promising cell therapeutics have also been developed to promote tissue regeneration, showing high potencies for the treatment of damaged liver. Nevertheless, in the remit of regenerative medicine, cell-therapy efficacies remain suboptimal as a consequence of the low engraftment of injected cells to the existing surrounding tissue. Herein, we present a facile approach to enhance the adhesion and engraftment of therapeutic hepatic progenitor cells (HPCs) through specific and homogeneous cell surface modification with exogenous polysaccharides, without requiring genetic modification. Coated HPCs exhibit significantly increased markers of adhesion and cell spreading and demonstrate preferential interactions with certain extra-cellular matrix proteins. Moreover, they display enhanced binding to endothelial cells and 3D liver microtissues. This translatable methodology shows promise for improving therapeutic cell engraftment, offering a potential alternative to liver transplantation in end-stage liver disease.

Liver transplantation currently stands as the only therapeutic option for multiple acquired and congenital liver diseases[1–4]. However, the paucity of liver donors limits this option to a minority of patients, highlighting a need for alternative treatment strategies[5–7]. Cell-based regenerative therapies may circumvent the requirement for organ donation and invasive organ transplantation through the ex vivo expansion and reintroduction of regenerative cells[8,9]. Pre-clinical in vivo models have consistently demonstrated remarkable levels of functional restoration at sites of engraftment within failing livers[10–12]. Despite these promising results, the challenge of low engraftment remains a significant hurdle, impeding the broader adoption of cell-based therapies in clinical practice[9]. Specifically, nearly 80–90% of transplanted hepatocytes are destroyed as a result of limited adhesion to the sinusoidal endothelium and other cytotoxic events, such as cytokine-mediated toxicity and oxidative stress[13]. In particular, hepatic progenitor cell (HPC) engraftment is even lower (<5%)[14], highlighting a clear need for exploring new methods to improve the overall efficacy of cell-based therapies.

Cellular engraftment is mediated by the cell membrane and the immediate pericellular matrix, making the cell surface a rational target for modification. Genetic modification (GM) strategies that alter the expression of surface receptors represent a traditional approach for site-specific homing of regenerative cells[15,16]. However, these methods are often time-consuming and costly, carrying substantial risks of insertional mutagenesis and adverse heterogenous expression[17–19]. As a result, alternative strategies have been developed to expand the plethora of foreign functionalities that can be conjugated to the cell surface, including bioactive and therapeutic payloads that can modify cell behaviour and cell interactions with the surrounding environment[20–25]. Among these, metabolic oligosaccharide engineering (MOE) pioneered by Bertozzi and co-workers has emerged as a promising technique for the integration of abiotic chemical functionalities onto cell surface glycans, which provide an ideal binding site for the attachment of macromolecules owing to their high abundance at the cell surface[26–30].

MOE employs unnatural sugar analogues bearing bio-orthogonal chemical handles to 'hijack' the cytosolic enzymes of the glycan biosynthetic pathway, effectively replacing naturally occurring sugars. The highest cell surface coverage can be achieved using $N$-acetyl mannosamine (ManNAc) derivatives, such as $N$-azidoacetylmannosamine-tetraacetate (Ac$_4$ManNAz), owing to their participation in the sialic acid biosynthetic

[1]School of Chemistry, University of Birmingham, Edgbaston, Birmingham, UK. [2]Healthcare Technologies Institute, Institute of Translational Medicine, University of Birmingham, Edgbaston, Birmingham, UK. [3]School of Chemical Engineering, University of Birmingham, Edgbaston, Birmingham, UK. [4]InSphero AG, Wagistrasse 27A, Schlieren, Switzerland. [5]Institute of Cancer and Genomic Sciences, University of Birmingham, Edgbaston, Birmingham, UK. ✉e-mail: a.elhaj@bham.ac.uk; m.c.arno@bham.ac.uk

pathway and efficient conversion to azidoacetyl sialic acids, which are incorporated into *N*- and *O*-linked sialoglycoconjugates[29–33]. Using this technology, the membrane of mammalian cells has been functionalized in vitro and in vivo through copper-free strain-promoted alkyne-azide cycloaddition (SPAAC) to introduce a wide range of functionalities, from fluorescent dyes for tumour targeting[34], peptides for immunomodulation[35], and synthetic polymers to provide long-lasting scaffold cellularization[36]. More recently, this chemistry has been employed for the formation of a protective layer that was found to suppress tumour growth in vivo[37], as well as for the development of living hybrid materials[36,38]. These recent studies provide relevant proof of concept for the application of MOE techniques in addressing healthcare challenges for cell-based therapeutics.

Herein, we functionalize regenerative HPC surfaces with polysaccharides, commonly used in tissue engineering applications owing to their biocompatibility and ease of modification[39,40]. In particular, we explore hyaluronic acid (HA) and alginate (Alg) as exogenous cell surface functionalities, owing to their ability to establish hydrogen bonds with extracellular matrix (ECM) proteins and glycans. These biopolymers have been previously exploited for bioadhesive and targeting applications in drug and cell-delivery[14,41–44]. We hypothesize that coating the cell membrane with these polysaccharides could enhance adhesion by facilitating hydrogen bonding at the endothelial interface, mimicking the rolling mechanisms observed in native carbohydrate-mediated interactions[45–47]. By covalently conjugating polysaccharide moieties through MOE and bio-orthogonal click chemistry approaches (Fig. 1a), we introduce a homogeneous, single-cell coating that successfully increases in vitro HPC adhesion to biologically relevant surfaces of ECM proteins and vascular endothelial cells. Additionally, we employ a microfluidic flow chip to investigate cell adhesion to hepatic microtissue models. Our findings reveal that HA-coated HPCs exhibit increased engraftment to 3D microtissues while maintaining progenitor cell potency, outlining a promising method for enhancing therapeutic cell adhesion in biomedical applications.

## Results

HA and Alg were functionalized with dibenzocyclooctyne-amine (DBCO-amine) through 4-(4,6-dimethoxy-1,3,5-triazin-2-yl)-4-methylmorpholinium (DMTMM)-mediated amidation[48,49]. Using this method, five and ten units of DBCO were conjugated to HA and Alg to obtain HA4, HA8 and Alg4, Alg8, with 4% and 8% degrees of functionalization as confirmed by $^1$H NMR spectroscopy (Supplementary Fig. 1–4). Subsequently, fluoresceinamine (FAM) was conjugated to all polysaccharide-DBCO derivatives to yield 5% functionalization, as quantified by UV-vis spectroscopy. After removal of unreacted reagents through dialysis, the final product was characterized by $^1$H NMR spectroscopy and SEC analysis, which showed the expected increase in molecular weights of HA and Alg derivatives after the cumulative conjugation of DBCO and FAM (Supplementary Fig. 5–10).

HPCs were selected for this study as they possess the advantage of ex vivo expansion and culture in comparison to primary hepatocytes, which require cryopreservation or immediate transplantation following harvesting[9–12]. After expansion, HPCs were incubated with 40 μM of Ac$_4$ManNAz to introduce an azide functionality at the cell surface. To optimize the level of azide groups on HPCs, cells were incubated with Ac$_4$ManNAz for 2 to 5 days and subsequently treated with Cy5-DBCO to allow visualization and quantification of the resultant fluorescence every 24 h through confocal fluorescence microscopy and flow cytometry. These data indicated that a maximal threshold of azide display was reached after four days of incubation with Ac$_4$ManNAz (Fig. 1b and Supplementary Fig. 11), with the fluorescence intensity not increasing further beyond this time point; hence a four-day incubation period was used in this study.

According to previous literature, in vitro click-mediated cell labelling has been predominantly carried out on monolayer cell cultures, under standard cell culture conditions of 37 °C, 5% CO$_2$[31–38]. To enhance translatability to a cell transplantation workflow and ensure a uniform whole-cell coating on individual cells (coating homogeneity), we aimed to perform click-mediated coating directly on suspended HPCs. Our preliminary

attempts at surface functionalization of suspended cells at 37 °C showed significant non-specific binding of Cy5-DBCO and DBCO-FAM-functionalized HA and Alg in both azide and non-azide treated cells, likely caused by non-specific membrane interactions and extensive internalization of the biopolymers whilst suspended in the coating media (Supplementary Fig. 12 and 13). These observations are in alignment with findings reported by Gibson and co-workers, where Cy3-DBCO and DBCO-functionalized poly(hydroxyethyl acrylamide)(pHEA)-FAM were also internalized by azide-treated human lung cancer fibroblasts[50]. For this reason, we employed a common approach used in molecular biology to reduce cellular internalization and explored incubation with the polysaccharides at 4 °C[51]. Indeed, this resulted in a significant decrease in non-specific binding and internalization, for both the small molecule Cy5-DBCO and the DBCO-FAM-functionalized biopolymers (Supplementary Fig. 12 and 13).

To determine the maximum viable concentration of DBCO polysaccharides to be used for coating, azide-treated HPCs were incubated with varying concentrations of HA-DBCO and Alg-DBCO (0-3 mg mL$^{-1}$) for 2.5 h and viability monitored over 72 h using a PrestoBlue proliferation assay (Supplementary Fig. 14). Both polysaccharides showed high cytocompatibility up to a concentration of 2 mg mL$^{-1}$. Interestingly, HA-DBCO-coated cells exhibited over 100% viability, likely as a consequence of the proliferative effect of HA, previously reported in the literature[52–54].

To determine the optimal incubation time with the polysaccharide coating solutions, HPCs were incubated at 4 °C under shaking with 1.5 mg mL$^{-1}$ of either functionalized HA or Alg, followed by flow cytometry analysis to quantify polymer conjugation (Fig. 1c, d). As expected, a slight increase in fluorescence was detected by flow cytometry for cells coated with HA8 and Alg8 compared to HA4 and Alg4, respectively accounting for more polymer conjugation at the cell surface with increasing DBCO functionalization (Supplementary Fig. 15). Significant increase in fluorescence was observed for both polymers when incubation time was increased from 1 h to 2.5 h (Fig. 1c, d). However, confocal fluorescence microscopy images revealed increased intracellular fluorescence after 2.5 h of incubation at 4 °C, likely as a result of polymer internalization over time (Supplementary Fig. 16). As such, an incubation period of 2 h was preferred and adopted in all our experiments. Incubation of non-azide bearing HPCs with DBCO-polysaccharides showed minimal non-specific binding, highlighting the specificity of such bio-orthogonal reactions (Supplementary Fig. 13). Confocal fluorescence microscopy images of Ac$_4$ManNAz-treated HPCs incubated for 2 h with HA (Fig. 1e) and Alg (Fig. 1f) derivatives demonstrate that homogeneous, whole-cell encapsulation could be achieved within this timeframe.

One critical advantage of MOE in the context of cell-based therapies is the covalent linkage of payloads throughout the cell surface. This offers the possibility to shed the polymer coating from the cell membrane in a physiological process, allowing de-coated cells to establish new interactions with the ECM and surrounding tissue in situ. The fluorescence intensity at the cell surface started to decrease after 24 h of incubation at 37 °C, 5% CO$_2$, when cells were left to adhere on a collagen-coated dish. Complete disappearance of the fluorescence signal could be observed after 96 h, as evidenced by confocal fluorescence microscopy (Fig. 2a, b, and Supplementary Figs. 17 and 18) and flow cytometry (Fig. 2c), with a level of quantified fluorescence comparable to uncoated cells at the end of this time window. As expected, no difference was observed between the two polysaccharide coatings or among the different degrees of DBCO functionalization. Internalization of the biopolymers was observed by analyzing intracellular slices of confocal Z-stack images, with intracellular fluorescence decreasing significantly after 24 h (Supplementary Figs. 19 and 20). Based on these data and previous reports, it is likely that the glycoconjugate anchors are being internalized and processed within the cell at a rate that is solely dependent on the specific membrane turnover of each cell population[50,55–57].

Integrins are the principal receptors in cells binding to tissues and play a role in engraftment of regenerative cells to the liver, owing to their ability to bind to a diverse range of ligands in the liver

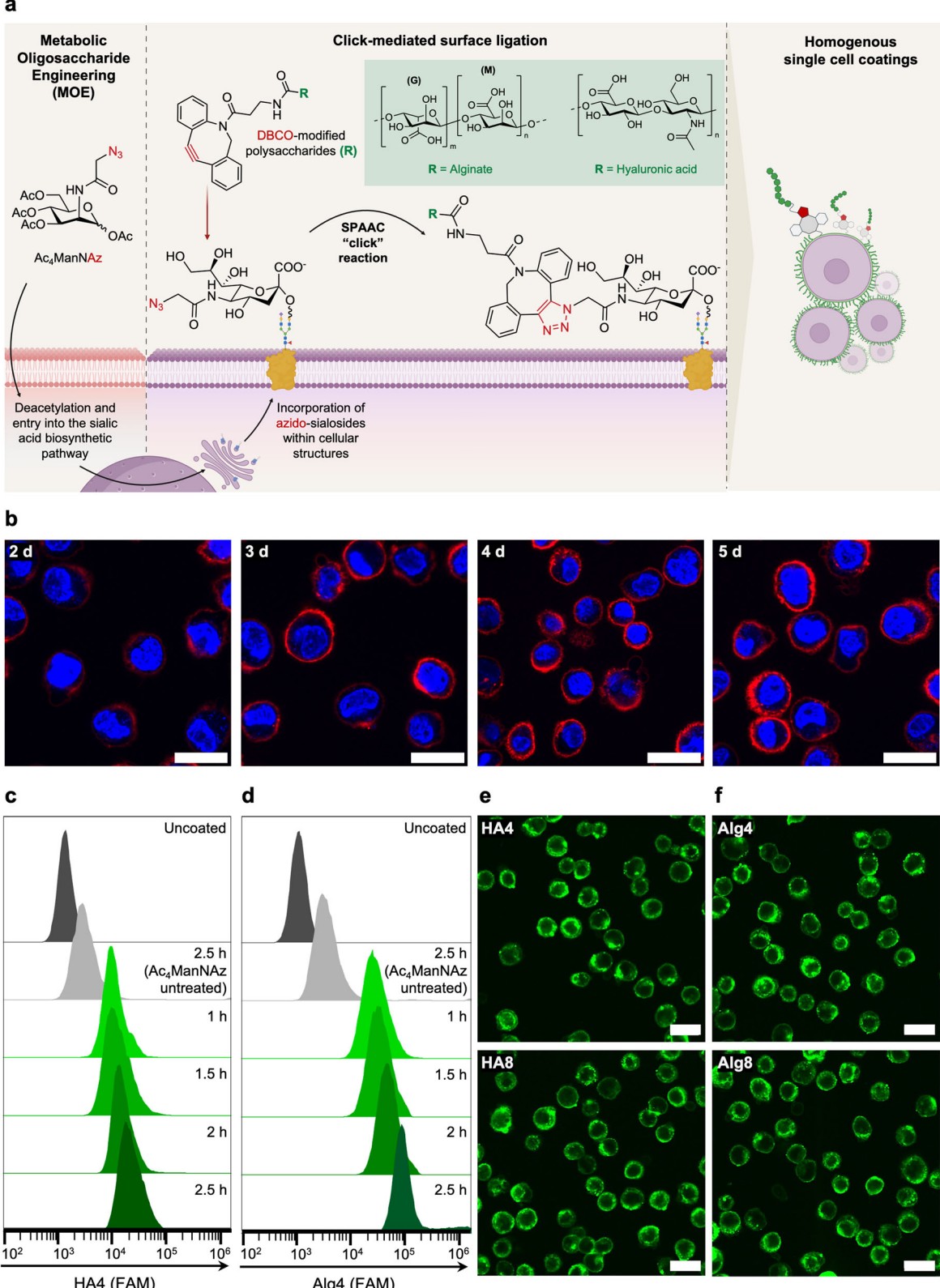

**Fig. 1 | Whole-cell polysaccharide coating of HPCs through MOE. a** Schematic of metabolic glycan labelling of mammalian cells resulting in the incorporation of *N*-azidoacetyl-D-neuraminic acid (NeuAz) on surface sialosides. Supplementation of dibenzocyclooctyne (DBCO)-modified polysaccharides results in a spontaneous strain-promoted alkyne-azide cycloaddition (SPAAC) click reaction, achieving a homogenous coating at the cell surface. **b** Confocal fluorescence images visualizing cell surface azides *via* conjugation with Cy5-DBCO (at 4 °C), after varying the incubation length of HPCs with Ac$_4$ManNAz from 2-5 days. Scale bar = 20 µm.

**c, d** Flow cytometry graphs of uncoated (dark grey), Ac$_4$ManNAz-untreated (light grey) and Ac$_4$ManNAz-treated HPCs (green) incubated with Alg4 and HA4 for the indicated time periods (1–2.5 h). "Uncoated" refers to Ac$_4$ManNAz-treated HPCs not incubated with polysaccharides, while "Ac$_4$ManNAz untreated" refers to HPCs incubated with polysaccharides without being previously exposed to Ac$_4$ManNAz. Confocal fluorescence images of HPCs treated with Ac$_4$MAnNAz for 4 days and subsequently incubated with HA (**e**) and Alg (**f**) with different degrees of DBCO functionalization (4% and 8%) for 2 h at 4 °C. Scale bar = 50 µm.

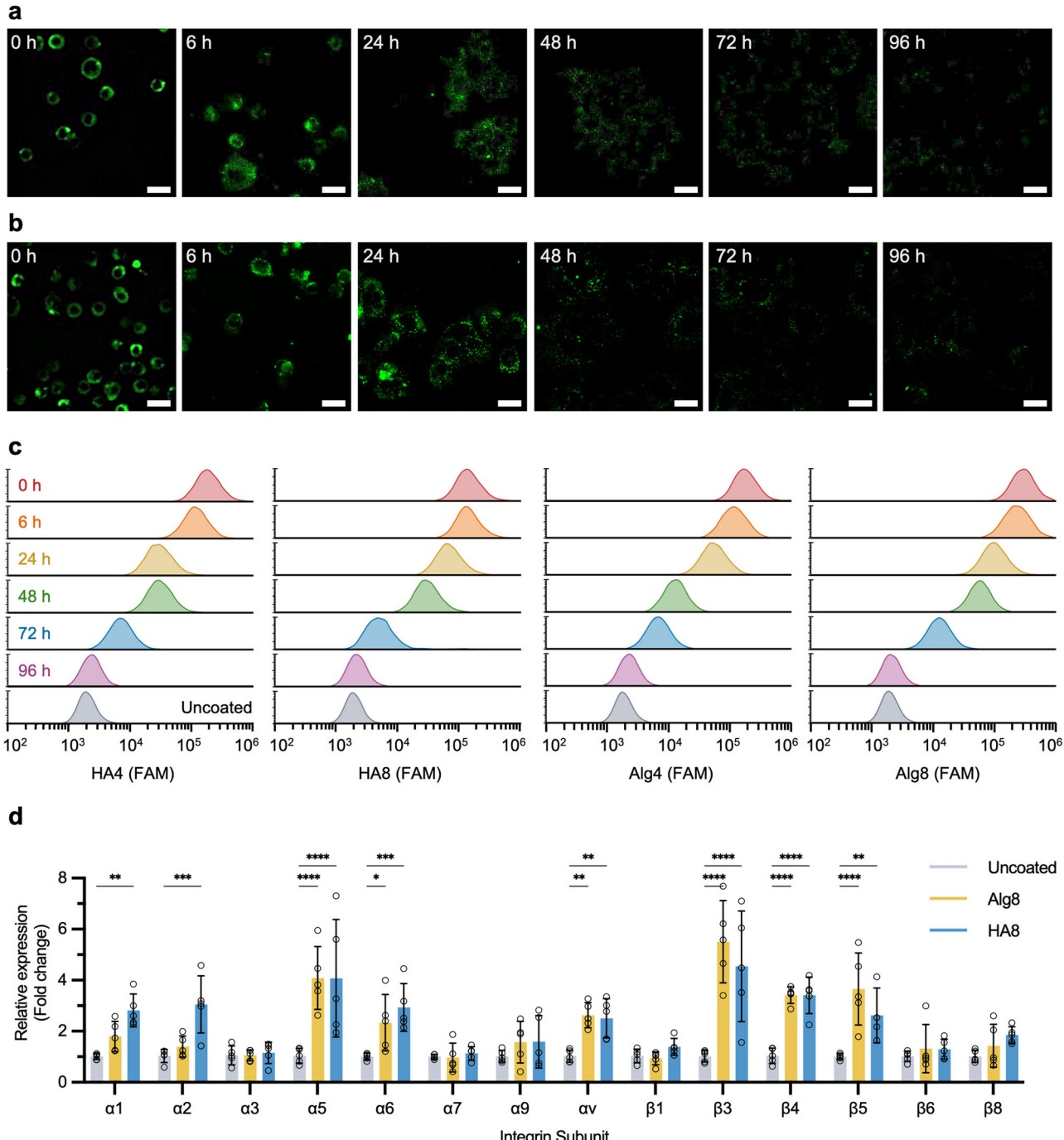

**Fig. 2 | Transient surface lifetimes of conjugated polysaccharides and gene expression of integrin subunits after coating.** Confocal fluorescence microscopy images of HPCs in standard cell culture media after coating with HA8 (**a**) and Alg8 (**b**) captured over a period of 96 h. Scale bar = 20 µm. **c** Flow cytometry graphs of HPCs coated with HA and Alg at different degrees of DBCO functionalization, showing a decrease in fluorescence over time. Cells were incubated at standard culture conditions post-coating (37 °C, 5% $CO_2$) and cell populations were treated with accutase prior to flow cytometry measurements. **d** Relative expression level of different integrin subunits quantified by PCR ($N = 5 \pm$ SD, * $P = 0.05$, **$P = < 0.01$, ***$P = < 0.001$ ****$P = < 0.0001$).

$ECM$[58]. To characterise the potential effect of coatings on integrin expression we assessed mRNA expression for an array of integrins (Fig. 2d). Integrin expression did not decrease 20 h after coating and in the case of integrins α1, α2, α5, α6, αv, and β3, β4, β5 an increase in expression was observed (full methods available in the Methods section). This indirect upregulation of integrins could potentially play a beneficial role in facilitating and maintaining enhanced engraftment.

In order to test our hypothesis and assess whether the polysaccharide coating was indeed able to increase cell adhesion, a colorimetric ECM adhesion assay was performed as an indirect measure of cell adhesion to an array of ECM proteins[59]. A significant increase in adhesion to multiple ECM proteins and glycoproteins was observed for HA-DBCO derivatives, up to 92% higher compared to uncoated cells (Fig. 3a). The differences in the HA interactions observed between the various collagen (Col) subtypes is likely a consequence of the intrinsic differences in isoform structure and

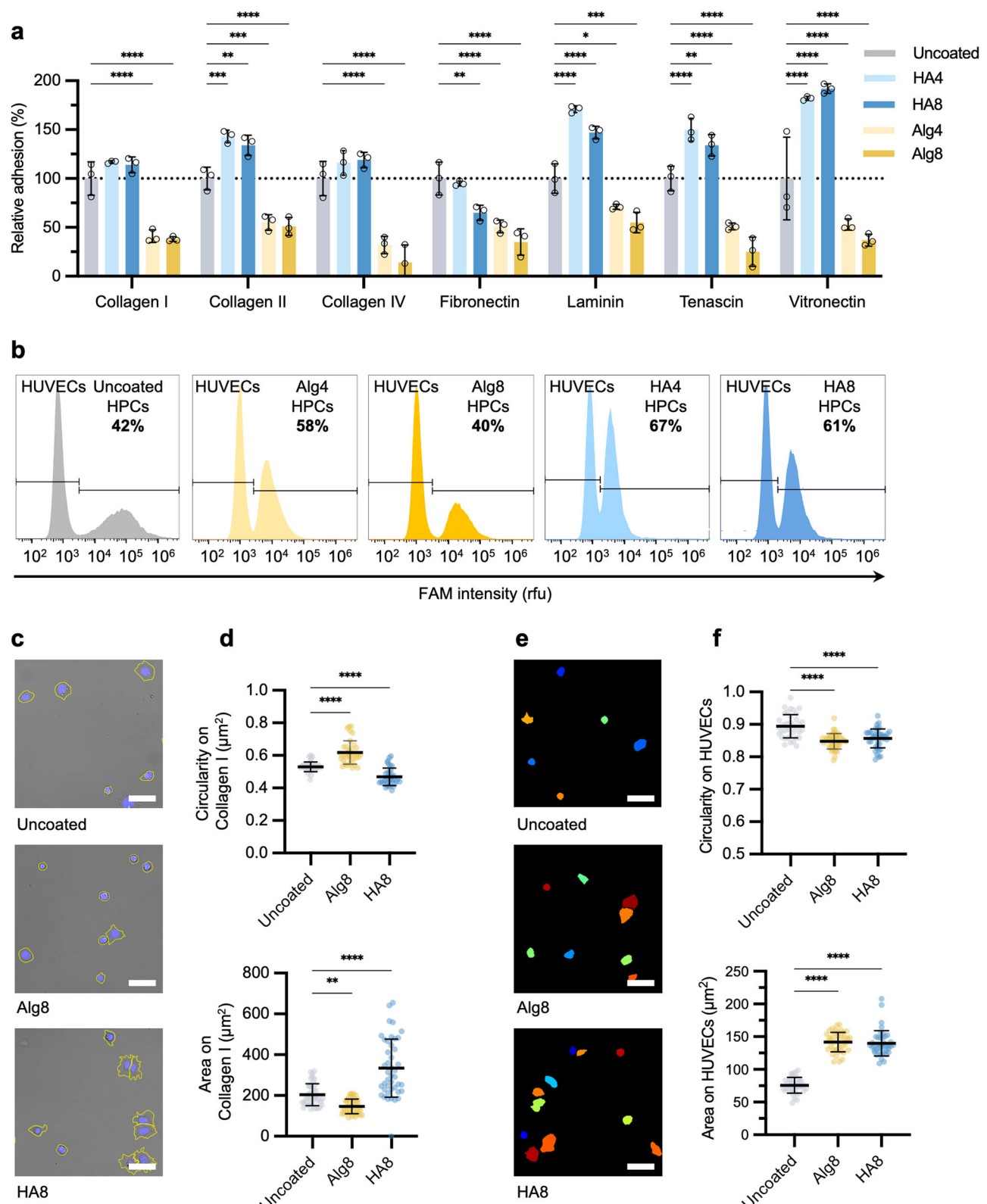

architecture[60]. For example, Col-IV is commonly found in basement membranes and naturally forms sheet-like structures rather than fibrils, thus not natively binding or interacting with HA. In contrast, Col-II forms a looser, mesh-like structure, which may account for increased hydrogen bonding interactions from exposed hydroxyl groups, in comparison to Col-I, which instead forms more densely packed fibrillar structures. Surprisingly,

no significant increase in HPC adhesion was found for fibronectin-coated surfaces, which contain specific HA binding motifs, suggesting that HPCs already natively bind at maximal levels to fibronectin. On the other hand, the significant increase in adhesion observed for laminin, natively located in basal lamina, is a likely consequence of the extensive post-translational glycosylation of laminin, which confers interactive hydrogen bonding sites

**Fig. 3 | Polysaccharide coatings modulate cell adhesion to ECM proteins and human umbilical vein endothelial cell (HUVEC) monolayers. a** Quantification of adhered HPCs coated with Alg and HA after adhesion onto ECM-coated 96-well surfaces. Data are presented relative to uncoated HPCs ($N = 3 \pm$ SD, **$P = < 0.01$, ****$P = < 0.0001$). The dotted line represents the mean adhesion of uncoated HPCs to ECM proteins. **b** Flow cytometry graphs of uncoated HPCs labelled with green fluorescent protein or coated HPCs after adhesion onto HUVEC monolayers for 2 h. Unbound cells were removed by PBS washings, and the relative proportions of HPCs adhered to HUVECs were quantified by flow cytometry after subsequent dissociation with accutase. **c** Confocal brightfield images of uncoated, Alg8, and HA8-coated HPCs left to adhere for 2 h on Col-I coated dishes. Yellow outlines indicate cell perimeters; nuclei are shown in blue. Scale bar = 50 μm. **d** Morphological parameters for area and circularity analyzed from confocal brightfield images for uncoated, Alg8, and HA8 coated HPCs after adhesion onto Col-I coated plates ($N = 3 \pm$ SD, **$P = < 0.01$, ****$P = < 0.0001$). **e** Confocal image masks of uncoated, Alg8, and HA8 coated HPCs left to adhere for 2 h on HUVEC monolayers. Scale bar = 50 μm. **f** Morphological parameters for area and circularity analyzed from image masks for uncoated, Alg8, and HA8 coated HPCs after adhesion onto HUVEC monolayers ($N = 3 \pm$ SD, ****$P = < 0.0001$). All adhesion experiments were performed under standard culture conditions post-coating (37 °C, 5% $CO_2$).

for HA compared to collagen[61]. Similar secondary interactions may be attributed to the increased adhesion levels of HA-coated cells for tenascin and vitronectin proteoglycans[62,63]. In contrast, HPCs coated with Alg4 and Alg8 showed significantly reduced binding to all ECM surfaces (Fig. 3a). This might be explained by the intrinsic capability of alginate to form ionic crosslinks with divalent cations such as $Mg^{2+}$ and $Ca^{2+}$, present in the cell culture media[64,65]. This ionic crosslinking may result in the formation of a relatively stiffer layer of alginate at the cell surface, effectively immobilizing the cell in a thin, encapsulating hydrogel. While hyaluronic acid is also able to crosslink with divalent cations, the presence of one carboxylic acid group per repeating unit, compared to two carboxylic acid groups in the case of alginate, may account for less robust ionic crosslinking. To explore this, we analyzed the morphologies of HA and Alg-coated cells after 2 h of seeding onto Col-I tissue culture dishes (Fig. 3c, d, and Supplementary Fig. 21). Col-I is the predominant collagen isoform in the liver ECM, and hence we decided to focus our experiments on this substrate, despite Col-II and other ECM proteins also showing increased binding to HA-coated cells. Col-I abundance also increases significantly during the progression of liver fibrosis, making it an ideal substrate for studying cell adhesion to fibrotic tissue[66,67]. Image analysis of segmented cell areas allowed for the quantification of spreading area and cell circularity on Col-I-coated plates. HA-coated cells showed a significant increase in spreading area, with increased formation of membrane extensions, as indicated by lower values of circularity. On the other hand, Alg-coated HPCs exhibited significantly lower cell spreading and increased circularity, hence supporting the hypothesis for the ionic-based formation of an alginate layer on the cell surface.

To determine the ability of coated HPCs to still interact and adhere to other cells, we explored their behaviour on confluent monolayers of human umbilical vein endothelial cells (HUVECs). HUVECs were seeded onto collagen-coated plates to model cell-cell interactions of HPCs with the sinusoidal endothelium. Uncoated HPCs, fluorescently labelled with green fluorescence protein (GFP), as well as HPCs coated with Alg or HA, were seeded on top of HUVEC monolayers and left to adhere for 2 h, the minimum time needed to observe significant changes in morphology between groups. Following extensive PBS washing and dissociation, flow cytometry analysis was used to quantify the HPCs adhered to the HUVEC monolayers (Fig. 3b). HPCs coated with HA4 and HA8 exhibited the highest level of adhesion to HUVEC monolayers, with a measured ratio of HPCs to HUVECs of 67% and 61%, respectively, compared to the 42% ratio measured for uncoated HPCs. Interestingly, HPCs functionalized with Alg4 also showed increased adhesion compared to the uncoated control, with a population value of 58%. This was surprising, considering the previous results obtained with our ECM adhesion model showed much lower adhesion. This may be explained by the oversimplified nature of ECM adhesion models, which do not consider the active participation of partner cells in cell-cell interactions. Alg8 coatings in this case showed no significant increase in the number of adhered cells compared to uncoated controls (40%). Consistent with our previous results on Col-I substrates, the quantification of morphological parameters suggests that HPCs coated with both HA and Alg are able to spread significantly more on HUVEC monolayers compared to uncoated HPCs, indicating a higher level of cell-cell interactions and adhesion (Fig. 3e, f, and Supplementary Fig. 22).

To further expand the complexity of our adhesion models, we sought to introduce fluid flow and shear stress as physiologically relevant parameters

to our adhesion assays. Akura™ ImmuneFlow chips were used to study adhesion to 3D InSight™ human liver microtissues (hLMTs), mimicking the in vivo environment where cells travel through blood vessels before reaching the site of engraftment. Moreover, the hLMTs are constructed with a range of different cells, including human hepatocytes, Kupfer cells, and lymphatic endothelial cells, mimicking the hepatic microenvironment found in vivo[68]. In this microfluidic platform, HPCs remain in suspension and continuous flow, using gravity and repeated back-and-forth tilting of the chip (Fig. 4a). Adhesion of HPCs is tested as the cells pass the 3D microtissues loaded in special culturing compartments, rather than static 2D cell monolayers. Considering that our HA coating has shown the most promising adhesion to both ECM components and HUVEC monolayers, we sought to explore both HA4 and HA8-coated HPCs for their adhesion to hLMTs. Confocal fluorescence microscopy analysis of HPCs coated with HA4 and HA8 showed increased adhesion to the liver microtissues compared to uncoated HPCs (Fig. 4b). Moreover, quantification by flow cytometry showed a significant increase in adhesion for HA4 and HA8-coated HPCs when compared to uncoated cells (3.2 and 3.5-fold, respectively) after 9 h of incubation mimicking in vivo shear conditions (Fig. 4c).

The differentiation potential of coated HPCs was assessed as a critical aspect for their potential translation to therapy. HPCs were treated with Wnt3a, which has previously been shown to promote differentiation of progenitors into hepatocytes[11,69]. Following this, HPCs showed morphological changes characteristic of hepatocyte differentiation, including increased cell size and hexagonal shape (Supplementary Fig. 23). Furthermore, intracellular glycogen storage was confirmed by periodic acid-Schiff (PAS) staining, indicating functional differentiation (Fig. 4d). Albumin production, another key marker of hepatocyte function, was quantified to show a significant increase following differentiation, further supporting the acquisition of hepatocyte-like phenotypes (Fig. 4e). Crucially, these findings demonstrate that the surface coatings do not impair the differentiation potential of HPCs, preserving their ability to develop into functional hepatocytes. These results highlight the translatability of our approach and reinforce its potential for cell-based regenerative therapies.

## Discussion
In this study, we used metabolic oligosaccharide engineering and bio-orthogonal click chemistry to functionalize regenerative hepatic progenitor cells with polysaccharides, with the aim to control cell adhesion mechanisms through targeted cell membrane modification.

Polysaccharides such as HA and Alg have been widely used in tissue engineering and regenerative medicine, owing to their advantageous bio-physical properties and well-documented biocompatibility[39,40]. Indeed, HA and Alg have previously been used in literature as bioadhesives and co-injectable hydrogels for drug and cell delivery systems to aid localization and retention[14,41,42]. We hypothesize that the attachment of such polysaccharides to the surface of HPCs could facilitate non-covalent intermolecular interactions, such as electrostatic and hydrogen bonding, to carbohydrate and protein moieties on target endothelial surfaces. Building on this hypothesis, we envisioned the use of polysaccharides as a proof of concept for the development of bioadhesive coatings for cell-based therapies.

The transient nature of our surface modification represents a significant advantage for therapeutic applications. Previous in vivo studies have demonstrated that HPCs (and hepatocytes used for cell replacement

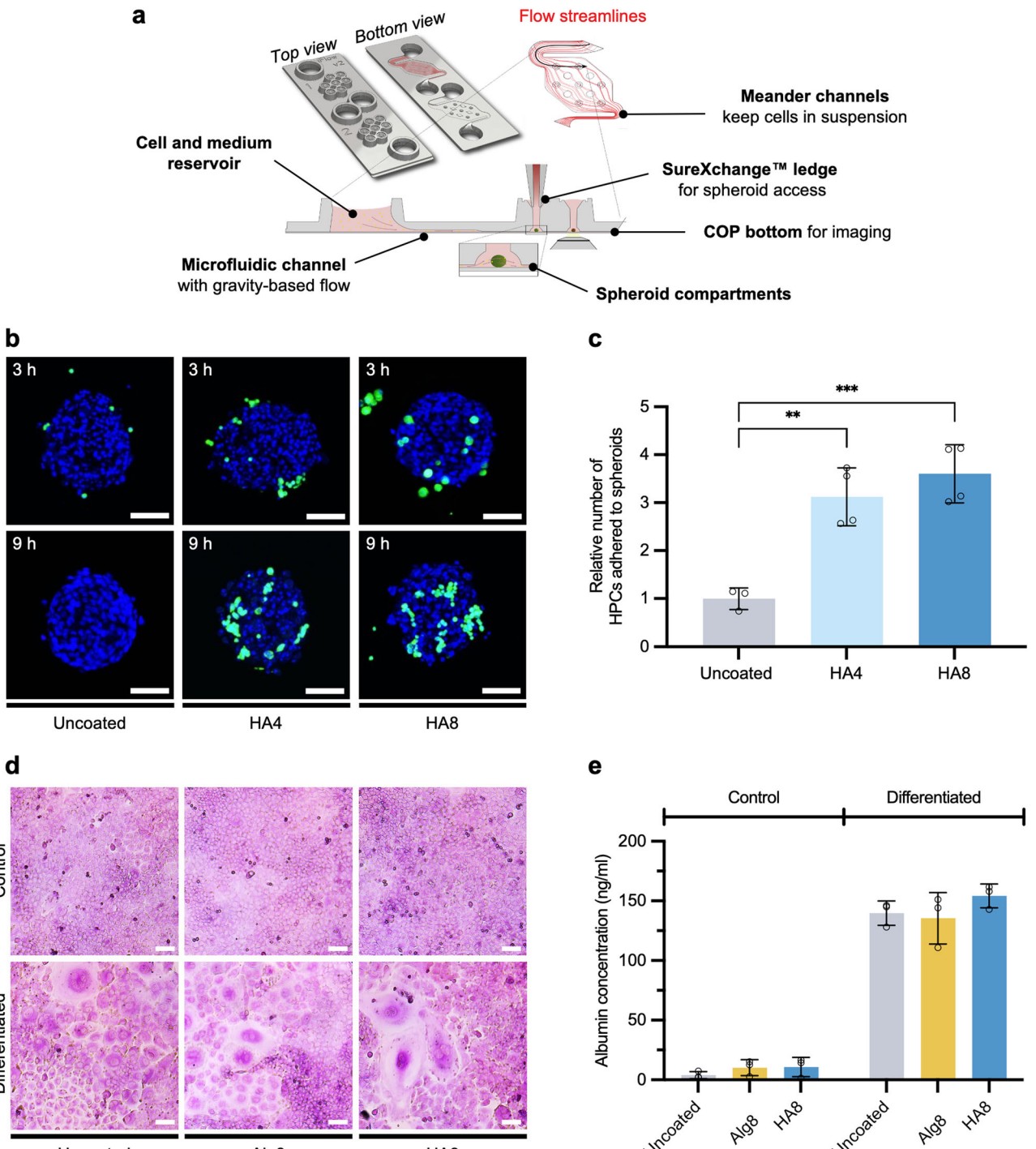

**Fig. 4 | Engraftment of coated HPCs to liver microtissues under microfluidic flow. a** Schematic representation of InSphero Akura™ ImmuneFlow chip used for the study of coated HPCs engraftment to hLMTs under gravity-induced flow. **b** Representative confocal fluorescence images of uncoated and HA-functionalized HPCs (green) engrafted to hLMTs after 3 h and 9 h of incubation under flow. Uncoated and coated HPCs were stained with CellTracker deep-red before injection in the microfluidic channels. Scale bar = 100 μm. **c** Cumulative flow cytometry quantification of uncoated, HA4, and HA8 coated HPCs engrafted to hLMTs after 9 h of incubation in flow (37 °C, 5% $CO_2$). Data are presented relative to uncoated HPCs ($N = 3 ±$ SD, ***$P = < 0.001$). **d** Brightfield images of PAS-stained HPCs after 6 days of incubation in media with (differentiated) or without (control) Wnt3a supplementation. Scale bar = 20 μm. **e** Murine albumin concentrations in cell culture media after differentiation with Wnt3a ($n = 3 ±$ SD).

therapy) can engraft within 1–4 days post transplantation[70–72]. This suggests that the critical window for polymer-assisted engraftment falls within these timescales. Indeed, the observed loss of fluorescence over 72–96 h aligns well with the temporal requirements for cell engraftment, promoting initial adhesion without long-term interference in other cell functions and processes.

After achieving single-cell surface coatings, we assessed adhesion to endothelial surface models of increasing complexity. Initial studies into the adhesion dynamics on Col-I and HUVECs revealed that HA-coated HPCs displayed increased morphological markers of adhesion, such as cell spreading and formation of cellular protrusions. This observation aligns with previous studies, whereby genetic overexpression of HA synthases

promoted the formation of membrane extensions and enhanced cellular adhesion to various physiological surfaces[73–75]. Converse to HA, Alg-coated cells displayed significantly reduced spreading and increased circularity compared to uncoated controls when seeded in both Col-I and HUVEC monolayers. This inhibition likely stems from the formation of a stiff alginate microlayer at the cell surface, formed as a result of intermolecular ionic crosslinking mediated by divalent cations present in the media.

The increase in integrin expression of subunits α1, α2, α5, α6, αv and β3, β4, β5, 20 h after coating suggests that the coatings may trigger a beneficial cellular response that could support long-term engraftment. The significant increase in integrin expression for both HA and Alg-coated cells may be ascribed to the higher stiffness sensed by cells as a result of the membrane coatings. This is consistent with previous studies that showed upregulation of integrins in response to stiff ECM surfaces[76,77]. Moreover, the presence of the relatively large, covalently linked biopolymers to *N*- and *O*-linked azido-sialosides may also indirectly increase sensitivity to mechanical forces, activating mechanotransduction axes such as the RhoA/ROCK and the YAP/TAZ pathways[78–80]. Notably, studies have demonstrated that forces as low as < 40 pN at the cell surface can trigger significant downstream mechanotransduction, including integrin activation[81,82]. It has been established that integrin expression is directly linked to their activation through cellular attachment. Inactive integrins are constitutively internalised and downregulated within 2 h of inactivity[83]. Activated integrins have been shown to have relatively long half-lives of 12–24 h owing to their attachment to surface ligands[84–86]. Therefore, the high level of sustained integrin expression in coated cells after 20 h suggests successful integrin activation. This is significant for our application, as it contributes to our goal of increasing cell engraftment post-transplantation; however, further studies are required to fully elucidate the mechanisms behind the observed upregulation.

Our investigations extend to showcase enhanced adhesion of HA-coated cells to 3D microtissue models of liver parenchyma under dynamic flow, mimicking the situation coated HPCs may encounter in vivo. A key limitation of this experiment is the species difference between the HPCs (mouse-derived) and the hLMTs (human-derived). Nonetheless, the adhesion observed is still biologically relevant, as also demonstrated in a recent study where similar human-derived biliary HPCs could engraft within mouse models of biliary disease, highlighting the potential for cross-species compatibility in such interactions[12].

A critical consideration for the clinical translation of this approach is its impact on cellular functionality. While our study demonstrates preserved HPC capacity to differentiate into hepatocytes, a further understanding of how cell surface modification affects critical cell functions will be crucial for ensuring that enhanced adhesion does not come at the cost of detrimental phenotypic changes. Additionally, it would be beneficial to investigate whether the temporary polysaccharide surface modifications can influence the immunological profile of modified cells, which could help in predicting potential host responses within clinical applications.

The findings of this study open possibilities for further modification of coatings and the exploitation of other biomolecules and bio-orthogonal chemistries for cell surface engineering[87,88]. For example, parameters such as molecular weight and architectures could be explored to further enhance molecular interactivity. Collagens, particularly type-I, could also be promising candidates as alternative cell coatings given their natural abundance in the liver and their role in hepatic progenitor cell niches[66,67]. Chitosan, with its polycationic nature and proven biocompatibility, might offer advantages for specific applications, particularly in situations where charge-based interactions could enhance adhesion and targeting[89].

The scalability and relative simplicity of our approach present significant advantages for potential clinical translation. The use of metabolic oligosaccharide engineering avoids the complexities and risks associated with genetic modification[18], while the chemistry involved is well-defined and amenable to GMP production, as evidenced by recent applications such as diagnostics, prodrugs, and antibody-drug conjugates[90–92], with the first 'click'-based therapy reaching clinical trials in recent years (NCT04106492)[93].

In conclusion, our work establishes a promising platform for enhancing therapeutic cell adhesion through surface engineering. The translatable nature of this approach allows for the modification of other therapeutic cell types. Future studies focusing on cellular functionality and in vivo validation will be crucial for realizing the full potential of this technology.

## Methods

### HA and Alg functionalization with DBCO and FAM

500 mg (11.1 μmol) of low-MW HA (Biosynth, FH01773) was dissolved in 10 mL of water with 8 eq. (24.6 mg, 88.8 μmol) of DMTMM (4-(4,6-dimethoxy-1,3,5-triazin-2-yl)-4-methylmorpholinium chloride) (Sigma-Aldrich, 74104). 6 Eq. (18.4 mg, 66.6 μmol) of dibenzocyclooctyne-amine (Sigma-Aldrich) dissolved in a 5 mL volume of tetrahydrofuran was slowly added to the solution and the reaction mixture was left to stir for 5 days to yield HA4. Molar equivalents were doubled to yield HA8. The product was then purified through dialysis for 3 days (MWCO 3.5 kDa) and lyophilized to yield a white product (87% and 93% yields for HA4 and H8, respectively). Alg4 and Alg8 were functionalized in a similar fashion, from a starting quantity of 500 mg (9.09 μmol) of low-MW Alginate (FMC biopolymers, Protanal LFR 5/60), yielding 85% and 88%, respectively. A 10 mg sample of DBCO-functionalized HA or Alg was subjected to a solubilization/lyophilization cycle in deuterium oxide ($D_2O$), three times. It was then redissolved in $D_2O$ for [1H] NMR spectroscopy analysis. Spectra were recorded on a Bruker 400 MHz spectrometer at 298 K and analyzed using MestReNova software. Each polysaccharide-DBCO variant was functionalized with 5-fluoresceinamine (FAM) (Sigma-Aldrich, 201626) by dissolving 200 mg of each polysaccharide variant (4.3 μmol, 4.2 μmol, 3.6 μmol, and 3.5 μmol for HA4, HA8, Alg4, Alg8, respectively) in 15 mL of 50:50 $H_2O$:EtOH, followed by the addition of 6 eq. of FAM (9 mg, 25.8 μmol for HA4, 8.8 mg, 25.2 μmol for HA8, 7.5 mg, 21.6 μmol for Alg4, and 7.3 mg, 21 μmol for Alg8). FAM quantification was performed by obtaining a UV-vis standard calibration curve of FAM at 488 nm in a 50:50 mixture of water and ethanol. The UV-vis absorption of 0.05% w/v of FAM-functionalized polysaccharides was measured, and the FAM content of the solution was subsequently extrapolated using Beer's law.

### NMR spectroscopy

[1H] NMR spectra were recorded at 400 MHz on a Bruker DPX-400 spectrometer in $D_2O$, unless otherwise stated. Chemical shifts are reported as δ in parts per million (ppm) downfield from the internal standard trimethylsilane.

NMR spectroscopy assignments for Alg-DBCO: [1H] NMR (400 MHz, $D_2O$) δ/ppm: 7.71-7.40 (8H, m, aromatic C*H* from DBCO), 5.06 (1H, s, C*H* from Alg), 4.51-3.48 (6H, m, C*H* from Alg); for Alg-DBCO-FAM: [1H] NMR (400 MHz, $D_2O$) δ/ppm: 8.81-6.88 ((8H, m, aromatic C*H* from DBCO) (9H, m, aromatic C*H* from FAM)), 5.07 (1H, s, C*H* from Alg), 4.50-3.50 (6H, m, C*H* from Alg); for HA-DBCO: [1H] NMR (400 MHz, $D_2O$) δ/ppm: 7.72-7.38 ((8H, m, aromatic C*H* from DBCO), 4.58-3.36 (8H, m, C*H* from HA), 2.02 (3H, m, C*H₃* from HA); for HA-DBCO-FAM: [1H] NMR (400 MHz, $D_2O$) δ/ppm: 7.72-7.38 ((8H, m, aromatic C*H* from DBCO), (9H, m, aromatic C*H* from FAM), 4.58-3.36 (8 H, m, C*H* from HA), 2.02 (3H, m, C*H₃* from HA).

### Size exclusion chromatography

Aqueous SEC measurements were performed on an Agilent 1260 Infinity II Multi-Detector GPC/SEC System fitted with RI and ultraviolet (UV, λ = 309 nm) detectors, using a 80:20 $H_2O$:MeOH elution solvent containing 0.1 M $NaNO_3$. Polymers were eluted through an Agilent guard column (PLGel 5 μM, 50 × 7.5 mm) and two Agilent mixed-C columns (PL aquagel-OH MIXED-H 8 μM, 300 × 7.5 mm) with a flow rate of 1 mL min$^{-1}$, 40 °C. $M_n$, $M_w$ and Đ were determined using Agilent GPC/SEC software (vA.02.01) against a 15-point calibration curve ($M_p$ = 615 - 3,187,000 g mol$^{-1}$) based on PEG standards (Easivial PEG-M/H, Agilent).

## HPC culture and general maintenance

HPCs were provided as a kind gift from Dr Wei-Yu Lu (University of Edinburgh). A step-by step protocol describing the method for primary cell harvest, isolation and maintenance has been previously described[11,94]. Briefly, T75 flasks were coated overnight with the addition of 5 mL of 200 µg mL$^{-1}$ A1110501 Col-I (Sigma-Aldrich, C3867). When coating well plates and 35 mm dishes, 1 mL and 0.5 mL of Col-I solutions were added to each well in 12 and 24-well plates, respectively. The Col-I solutions were removed and washed with fresh PBS before use, with a working volume for the 35 mm dishes of 2 mL. General maintenance of the cell line was completed by passaging every 5-7 days or before reaching 90% confluency. Cells were dissociated using Accutase cell dissociation reagent (Gibco, A1110501) and re-seeded at a density of $3 \times 10^5$ cells per T75 cell culture flask. HPCs were cultured in complete DMEM supplemented with 10% FBS, 2.0 mM L-glutamine, and antibiotic solution containing penicillin (100 units mL$^{-1}$) and streptomycin (100 µg mL$^{-1}$) at 37 °C in a humidified atmosphere containing 5% $CO_2$. Cells were counted by standard trypan blue exclusion using an automatic cell counter (Countess II, ThermoFisher).

## Cell surface functionalization and analysis

Prior to incubation with the polysaccharide solution, dissociated HPCs were seeded into fresh Col-I treated T75 flasks at a cell density of $3 \times 10^5$ cells per T75 cell culture flask (full HPC culture and maintenance methods detailed in the section above). Standard DMEM solution supplemented with 40 µM Ac$_4$ManNAz was added to cells, which were then left to proliferate for four days prior to coating at 37 °C in a humidified atmosphere containing 5% $CO_2$. DBCO-FAM functionalized HA or Alg was then dissolved in PBS, by heating to 37 °C and mixing in a vortex mixer (3000 rpm, 1 min.) to obtain a 0.75% w/v stock solution. HPCs were detached using accutase (Gibco, A1110501) cell dissociation solution to yield a cell suspension in DMEM. 1.5 $\times 10^6$ Cells were resuspended 1.8 mL DMEM supplemented with a final concentration of 0.15% w/v from the polysaccharide stock solution. Cells were left to incubate in a cooled shaking incubator for 2 h (4 °C, 80 rpm), followed by washing with PBS ($3 \times 1$ mL) and filtration using 40 µm cell strainers (3×). Cell suspensions were then either placed in 35 mm confocal dishes for confocal fluorescence microscopy analysis, adhesion analysis, or flow cytometry characterization.

## Confocal fluorescence microscopy

All cell images were captured by live cell confocal microscopy using an Olympus FLUOVIEW Spectral FV3000 laser scanning microscope equipped with 405, 488, 561, and 640 nm lasers. Olympus cellSens software was used for processing confocal microscopy data. For live cell experiments, cells were kept under controlled conditions of 37 °C, 5% $CO_2$.

## Flow cytometry

To quantify click-mediated surface functionality, flow cytometry of suspended cells was performed on a Beckman Coulter CytoFLEX flow cytometer with 4-lasers capable of 15 parameter analysis including FSC and SSC. Sample analysis required the use of the 488 nm excitation laser and a 530 nm filter for fluorescein/GFP measurements and 638 nm excitation laser and 660 filter for Cy5/Celltracker deep red measurements. All sample measurements consisted of a minimum of 30,000 total recorded events. Cells were suspended in PBS solution supplemented with 3% foetal bovine serum (FBS) and 3 mM EDTA and passed through a 40 µm cell strainer to ensure single cell analysis. Voltage settings applied ensured that untreated control cells appeared at low fluorescence emission intensities (FITC and Cy5 channels) and to ensure fluorescence measurements were within the detection range (<10$^6$ A.U.). CytExpert software (Beckman Coulter) was used for data collection. Conventional gates for removal of cell debris and doublet discrimination were applied to all samples ie. FSC/SSC plots, followed by quantification of coating/celltracker Deepred fluorescence (FITC or Cy5 channels, respectively, Supplementary Fig. 24). FlowJo was used for data presentation.

## Viability assays and quantification of HA and Alg lifetimes

Viability of cells incubated with non-fluorescent DBCO-functionalized polysaccharides was quantified by seeding HPCs onto Col-I coated 24-well plates seeded with $3.8 \times 10^4$ cells per well, followed by the addition of PrestoBlue cell viability reagent (Invitrogen, A13261) at each time point (24–72 h), following manufacturer's protocol. Endpoint fluorescence measurements at 590 nm were taken with a FLUOstar Omega microplate reader (BMG Labtech) using Omega MARS software (BMG Labtech).

The lifetime of DBCO-FAM polysaccharides coating on HPCs was quantified by seeding the cells on Col-I treated 35 mm confocal dishes with 1 $\times 10^5$ coated or uncoated cells to achieve 20% confluency. At each time point considered (6–96 h), HPCs were washed with PBS ($3 \times 1$ mL) followed by live cell confocal fluorescence microscopy imaging. At least five Z-stack images (imaging depth of 20 µm) from each dish ($N = 3$) were captured at each time point to monitor the presence of HA or Alg. Average cell fluorescence intensities were quantified using Olympus CellSens software in which a set value for image thresholds was applied to all images in the FAM channel, followed by ROI selection of cells and fluorescence intensity quantification. Fluorescence intensities of coated cells were also quantified by flow cytometry at each time point, following treatment with accutase cell dissociation reagent.

## RNA isolation, cDNA synthesis and qPCR

Coated cells were seeded onto collagen-coated plates with culture media and maintained in the incubator at 37 °C and 5% $CO_2$. After 20 h in culture, cells were detached using 0.5% Trypsin-EDTA (Gibco) followed by media to inhibit the trypsin and harvested by centrifugation for 5 min. at 200 g. Cells were resuspended in TRIzol Reagent (Invitrogen) immediately after collection and stored at −20 °C before total RNA extraction. Extraction of total RNA was performed according to the TRIzol manufacturer's protocol. RNA was diluted with nuclease-free water, digested with DNase I (Invitrogen) and then stored at −80 °C until further use. RNA concentration of each sample was assessed using a Spark spectrometer and NanoQuant plate (TECAN). Subsequently, total RNA was reverse transcribed to synthesize complementary DNA using SuperScript IV Reverse Transcriptase (Invitrogen), according to manufacturer's protocol. Real-time quantitative PCR was performed using SYBR Green PCR Master Mix (Applied Biosystems) according to the manufacturer's instructions on a AriaMx instrument (Agilent). PCR conditions used were as follows: initial polymerase activation at 95 °C for 15 min. followed by 40 cycles of denaturation at 94 °C for 15 seconds, annealing at 55 °C for 30 seconds, and extension at 72 °C for 30 seconds. The following commercial QuantiTect primers (QIAGEN) were used: Gapdh (QT01658692), integrin alpha 1 (QT01198554), integrin alpha 2 (QT01540798), integrin alpha 3 (QT00125678), integrin alpha 5 (QT00114611), integrin alpha 6 (QT00144354), integrin alpha 7 (QT00136990), integrin alpha 9 (QT00172459), integrin alpha v (QT00095235), integrin beta 1 (QT00155855), integrin beta 3 (QT00128849), integrin beta 4 (QT00269010), integrin beta 5 (QT00108976), integrin beta 6 (QT00128233), integrin beta 8 (QT00280686). Gene expression data was based on Cq values and quantified using the relative quantification $2^{-\Delta\Delta CT}$ method. Gene expression levels of the target transcripts were normalized against housekeeping gene GAPDH. Results were reported as mean of fold change relative to uncoated HPCs.

## Quantification of HPC adhesion to ECM and HUVECs

HPCs were coated with non-fluorescent DBCO-functionalized HA or Alg as outlined above and seeded at $1 \times 10^5$ cells per well on a 96-well plate ECM adhesion array (Sigma-Aldrich, ECM540), following manufacturer's protocol and previous literature[95]. Cells were incubated for 1 h at 37 °C, 5% $CO_2$. After incubation, media was discarded and cells were washed with assay buffer ($3 \times 100$ µL), followed by 5 min. incubation at RT with cell stain solution and washed with PBS ($3 \times 100$ µL). Stained cells were left to air dry for 5 min., followed by treatment with extraction buffer and gentle shaking

on an orbital shaker for 10 min. Endpoint absorbance at 544 nm was taken with an Omega FLUOstar microplate reader and Omega MARS software (BMG Labtech).

HUVECs (Gibco, C01510C) were cultured as standard in Human Large Vessel Endothelial Cell Basal Medium (Gibco, M200500) supplemented with low serum growth supplement (Gibco, S00310). Cell maintenance was carried out by replacement of medium every two days and passaging upon reaching around 60% confluency by dissociation with trypsin-EDTA solution. To obtain HUVEC monolayers, cells were seeded on either 6-well plates or 35 mm confocal microscope dishes pre-coated with Col-I at a cell density of $5 \times 10^4$ cells cm$^{-2}$. At 100% HUVEC confluency, $1 \times 10^6$ coated or uncoated GFP$^+$ HPCs were seeded onto the HUVEC monolayers and left to incubate for 2 h, followed by washing with PBS ($3 \times 1$ mL). The resulting co-culture was then analyzed by flow cytometry. Flow cytometry samples required prior dissociation into a single cell suspension by treatment with accutase cell dissociation reagent and resuspension in PBS flow buffer supplemented with 3% v/v FBS and 3% v/v EDTA. Cells were then filtered using a 40 μm cell strainer.

### Cell morphology characterization

HPCs were coated with HA or Alg as described above and seeded onto collagen-coated 35 mm confocal dishes at a seeding density of $5 \times 10^5$ cells per dish. Cells were left to adhere for 2 h and then washed with PBS ($3 \times 1$ mL) to remove non-adherent cells. Nuclei were then stained with 1 μM Hoechst 33342 for 15 min. Confocal laser scanning microscopy was used to capture a minimum of 12 images from each dish. Morphological parameters were quantified by outlining cell perimeters as regions of interest and using the measure feature in the brightfield (FIJI). At least 700 cells per experiment were measured.

For morphological analysis of cells adhered to HUVECs, 35 mm confocal microscope dishes containing 100% confluent monolayers of HUVECs were prepared as described before. Prior to seeding, HPCs (coated or uncoated) were stained with Celltracker DeepRed dye, following manufacturer protocol (1 μM, 10 min.). $5 \times 10^5$ Cells were seeded per dish and left to adhere for 2 h, after which unbound cells were washed with PBS ($3 \times 1$ mL). Nuclei were stained with 1 μM Hoechst 33342 for 15 min. A minimum of 12 confocal laser scanning microscope images were then captured from each dish followed by CellProfiler analysis of the Cy5 fluorescence channel to quantify HPC morphology.

### Adhesion assays using Akura™ ImmuneFlow chips

Akura™ ImmuneFlow chips (https://insphero.com/3d-cell-culture-tools/akura-immune-flow-organ-on-chip-platform/) were prepared according to manufacturer's protocol. hLMTs (InSphero, MT-02-302-04) were loaded into the microtissue compartment, followed by the addition of uncoated and coated HPCs (labelled with CellTracker deep-red) in a 200 μL DMEM suspension at a cell density of $50 \times 10^3$ cells mL$^{-1}$. The microfluidic chips were loaded onto Akura™ All-in-one Tilting System (InSphero) following a tilting programme of: $\pm$ 85° tilt (25 s movement, 1:15 min hold), horizontal pause (5 min.). hLMTs were imaged at time points of 3 h and 9 h. At the 9 h time point, PBS was flowed through the chips to remove unbound suspension of cells. hLMTs were collected and dissociated by the addition of 200 μL accutase with gentle vortexing for 30 min. The dissociation of hLMTs was routinely monitored using an inverted light microscope. Following complete dissociation, single cell suspensions were analyzed by flow cytometry, quantifying for CellTracker deep-red labelled HPCs. The experiment was repeated a minimum of 3 times.

### In vitro differentiation of HPCs

Cell differentiation was performed as previously described in literature[11]. Briefly, differentiation media composed of 100 ng mL$^{-1}$ of recombinant murine Wnt3a (Bio-techne) and 1% DMSO (Sigma-Aldrich) was added to William's E expansion media (Gibco) consisting of 10% FBS (Gibco), 17.6 mM NaHCO$_3$ (Sigma-Aldrich), 20 mM HEPES buffer (pH 7.5, Sigma-

Aldrich), 10 mM nicotinamide (Sigma-Aldrich), 1 mM sodium pyruvate (Sigma-Aldrich), 1X insulin, transferrin, selenium solution (ITS) (Gibco), 100 nM dexamethasone (Sigma-Aldrich), 0.2 mM ascorbic acid (Sigma-Aldrich), 14 mM glucose (Sigma-Aldrich), 10 ng mL$^{-1}$ IL-6 (Peprotech), 10 ng mL$^{-1}$ HGF (Peprotech), 10 ng mL$^{-1}$ EGF (Sigma-Aldrich). HA8- and Alg8-coated and uncoated HPCs were seeded on 24-well plates at $0.02 \times 10^6$ HPCs per well in triplicate. Cells were supplemented with 0.5 mL differentiation media or standard William's E media (without Wnt3a / DMSO) as non-differentiation controls. Cells were cultured in standard conditions of 37 °C, 5% CO$_2$ for 6 days, changing media after 3 days. Medium was stored at −80 °C for ELISA quantification of albumin using mouse albumin ELISA kit (Abcam) following manufacturer protocols. At day 6, cells were fixed using 4% v/v formaldehyde solution. Intracellular glycogen granules were visualized using the Periodic acid Schiff (PAS) staining kit following manufacturer protocols (Atom Scientific).

### Statistics and reproducibility

Group differences were examined with GraphPad Prism 8. Statistical comparisons of multiple samples were performed using a one or two-way analysis of variance (ANOVA), followed by Dunnett's multiple comparisons test. A $P$ value of less than 0.05 was considered statistically significant. Sample sizes and number of replicates, including whether independent samples ($n$) are analyzed or independent experiments ($N$) are performed, are indicated in each figure caption and legend.

### Reporting summary

Further information on research design is available in the Nature Portfolio Reporting Summary linked to this article.

## Data availability

All data generated or analyzed during this study are included in the published article and Supplementary Information. Source data for the main figures are provided in Supplementary Data 1 and all other data included in this study are available upon request by contacting the corresponding author.

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

## Acknowledgements

A.R.A. thanks EPSRC and SFI Centre for Doctoral Training in Engineered Tissues for Discovery, Industry and Medicine (grant number EP/ SO2347X/1) for support through a Ph.D. scholarship. EPSRC is thanked for supporting this project through the provision of a New Investigator Award to MCA (grant number EP/X037622/1). UK Research and Innovation (UKRI, Future Leaders Fellowship, MR/X033546/1) are thanked for supporting M.C.A. MRC UKRMP ECE Hub (# 623924) and MRC Exploiting In Silico Modelling To Address The Translational Bottleneck In Regenerative Medicine Safety (# 996932) are thanked for support to A.J.E.H. and M.R.V. ERC Advanced Award # 789119 is thanked for support to A.J.E.H. We acknowledge the support of the Technology Hub Facilities at the College of Medicine and Health, University of Birmingham, for providing access to flow cytometry and technical expertise. HPCs were provided as a kind gift from Dr. Wei-Yu Lu (University of Edinburgh). GFP+ HPCs were kindly donated by Dr Candice Ashmore-Harris and Prof. Stuart Forbes (University of Edinburgh). Figure 1a was created in BioRender. Alipio, A. (2025) https://BioRender.com/zbm9czf.

## Author contributions

M.C.A. and A.J.E.H. designed the study. T.H., L.H., and O.F. designed the Akura™ ImmuneFlow chip and trained A.R.A. in performing the experiments. A.R.A. performed the majority of the experiments and data analysis. M.R.V. and A.R.A. performed the PCR experiments and conducted data analysis on those. A.R.A., M.C.A. and A.J.E.H. wrote the manuscript with the contribution of all authors.

## Competing interests

The authors declare the following competing interests: InSphero AG has licensed rights to the Akura™ ImmuneFlow chip and hLMTs used in this study. All other authors declare no financial interests.
