## [Transparent Peer Review file · Communications Biology]

Sweet and sticky: increased cell adhesion through click-mediated functionalization of regenerative liver progenitor cells

Corresponding Author: Dr Maria Arno

Version 0:

Reviewer comments:

Reviewer #1

(Remarks to the Author)

The manuscript by Alipio et al. describes a method for covalently tethering functionalized polysaccharides of hyaluronic acid (HA) and alginate (Alg) to cell surfaces for homogeneous surface remodelling. Their work applies a commonly used metabolic oligosaccharide engineering (MOE) with Ac4ManNAz to engineer hepatic progenitor cells (HPCs) with SiaNAz on their surface. DBCO-functionalized fluorescent polymers of HA and Alg, prepared through amide ligation reactions, were then conjugated to the SiaNAz-HPC surface through SPAAC, creating HA- or Alg-engineered HPCs. The authors describe an initial optimization of MOE of HPCs, examining various MOE labeling times and detecting with a DBCO-Cy5, demonstrating that broad cell-surface remodelling was achieved following a 4-day Ac4ManNAz treatment. While the dose of DBCO-HA or Alg polymers for cellular ligation was not examined, the authors describe how conjugation time and temperature of the polysaccharides can alter labeling efficiency, ultimately using a 2 h conjugation time to the cell surface at 4 °C. The lifetime display of the polymers was examined by confocal microscopy and flow cytometry, showing a time-dependent decrease over 96 h. The authors also examined the polymer-coated HPCs' ability to adhere to a variety of ECM proteins, a HUVEC monolayer and liver-derived microtissues in a microfluidic flow-based assay, demonstrating that the HA-coated HPCs had improved adhesion and engraftment on liver microtissues.

Overall, the manuscript is easy to read and the importance of the work is justified. While the strategy of covalently ligating functional molecules to SiaNAz-engineered cells through SPAAC is not novel, the manuscript describes an important translatable application of this methodology. The work suggests that cell-surface remodelling of HPCs with polysaccharides is a promising method for enhancing cell adhesion that may have high potential for cell engraftment therapies. Cell-based regenerative therapies hold promise as alternative strategies to certain whole-organ transplantation. While transplanted cells can achieve functional restoration in failing organs/tissues, current treatments suffer from poor engraftment of cells at the transplant site. Thus, being able to functionalize cells with molecules that can improve engraftment is significant, and applying non-genetic techniques like MOE to achieve may be a viable strategy.

While it is evident that the addition of their polymer coating to HPCs, namely the HA-polymer, improves adhesion and engraftment on liver microtissues, there are several concerns that should be addressed to support the findings and conclusions of this work. Once addressed and revised, the manuscript would be suitable for publication and would be of interest to researchers in the glycobiology and regenerative medicine fields:

1. Controls for HA- or Alg-polymer treated HPCs alone (i.e. in the absence of MOE) are not provided, with the exception of data shown in Figure S12c/d for the 4°C SPAAC. On page 6 the authors state that at 37°C there is "extensive internalization of the biopolymers and non-specific membrane interactions even in non-azide treated cells", but this control data is not provided. Further, it does appear that some fluorescein signal in the HA8- and Alg8-treated polymer control samples at 4°C SPAAC is present, particularly for the Alg-polymer control (Figure S12c/d), suggesting there may be background or non-specific adherence of the polymers under these conditions. These controls are important to include in flow cytometry data in Fig. 1c-f to better quantify the background signal of the polymer. If significant background non-specific binding of the DBCO-polymers is observed, these polymer-alone controls (i.e. non-MOE treated HPCs) would be needed in the functional assays, such as the HUVEC adhesion assay in Fig. 3b, to demonstrate that the covalent attachment of the polymers to SiaNAz-engineered cells is required for the adhesion they are reporting.

2. The discussion and interpretation data of the polymer lifetime-display data (page 7 discussion; Data Fig. 2 and Fig S16) could be expanded and made more clear for how the authors are coming to their conclusions stated. It is suggested that the coatings “started to visibly detach from the surface after 24 h.” What evidence suggests that the coatings are visibly detaching? Is it known whether the polymer is being shed, or if the glycoconjugate anchors are being internalized and digested/recycled within the cell and are not present on the cell membrane? After 6 h, it appears that some internalization can be seen by confocal microscopy. At 24 h and beyond, the FAM signal appears quite punctate and it is unclear whether the signal is restricted to the cell surface or is intracellular. Analysis of the Z-stacks of the confocal microscopy data should help clarify whether the signal is intracellular or restricted to the surface. Additionally, the Fig. 2d caption for the flow cytometry data specifies it is cell-surface fluorescence being observed; however, if the polymers are internalized (e.g. present in lysosomes), but not degraded, would a fluorescence signal not still be observed for the internalized polymer? It would be helpful for the authors to clarify these points to support their conclusions. In the discussion section (page 14), the authors state that “the transient nature of the conjugated polysaccharide functionalities aligns well with the regenerative timescale.” What is this timescale? It would be useful for the authors to comment on what the desired lifetime of display of their polymer would be for their engraftment-enhancing functions and for future translation into a regenerative therapy (i.e. if their polymer is still present on the surface, albeit at a reduced amount due to turnover and internalization, is this timeline translatable?).

3. The authors characterize their functionalized polysaccharides using a combination of ¹H NMR spectrometry, UV-vis spectroscopy, and size exclusion chromatography. For the HA polymers, integration of ¹H NMR signals for DBCO relative to the acetamide singlet gives a reasonable estimate for DBCO functionalization, as has been done in similar literature (DOI: 10.1007/s10856-023-06757-9). However, without a clear acetamide peak for Alg, the overlapping signals from 3.7-4.2 ppm were for relative integration (integrating to 4 as the H2-H5 of the monosaccharide repeating unit); because of the broadening of signals due to the polymeric nature of Alg, this may not accurately quantify the DBCO functionalization. It would strengthen the authors claims to consider the following:

a. Does integration of H2-H6 region of the DBCO-HA (~3.2-4.0 ppm) align with the 4% and 8% DBCO functionalization quantified by the acetamide singlet integration (2 ppm)? If yes, this suggests the quantification of the DBCO-Alg polymers is accurate. Additionally, the signal in the aromatic region is very low in Figs. S5/7; more scans should be acquired to increase signal beyond background noise to confirm DBCO/FAM quantification.

b. DBCO absorbs at 290 and 310 nm? Can the authors use this absorbance to quantify DBCO-HA/Alg functionalization by UV-vis spectroscopy, similar to FAM quantification?

4. The methods are well written and fully describe the experimental protocols for the data provided. The manuscript would be strengthened by clarifying and including more details of the experiment conditions in the figure captions and in the figures. For example, given the differences in display of DBCO-HA/Alg depending on time and temperature of incubation, it should be clear in the figure and figure caption what time and temperature were used (Fig. 1b, S12). Additionally, the number of replicates, error bars, and details statistical analyses were not provided for some experiments (e.g. Fig. 3b, 4c). Were these experiments replicated? The authors should expand figure captions and labels within figures to ensure that all relevant experimental details and data are included.

5. The sialic acid structure in Fig. 1a should be corrected. The authors should ensure the correct stereochemistry is displayed at C2 and C5, and the “Ac” on the exocyclic N should be “H”. In MOE literature, the chemical structures of sugars like ManNAz and sialosides are commonly drawn in a chair conformation (see Fig. 2C in DOI: 10.1042/BCJ20200612 for an example). The Fig. 1 caption should be corrected to reflect that NeuAz is incorporated on glycan structures in lieu of sialic acid residues, not on Sia-containing residues.

Other comments:

1. The authors could consider providing additional details in the introduction to expand on the rationale for remodelling cell surfaces with Alg and HA for engraftment. It is noted that Alg and HA can interact with ECM proteins and free glycans, but more mechanistic context for their role in maintaining/regulating cell adhesion would be helpful. It would also be useful to include the general repeating structures of HA/Alg in Figure 1.

2. Page 2, Ref 12 suggests low engraftment is due to more than inadequate adhesion to the sinusoidal epithelium (oxidative stress, immune activation, etc). For accuracy, the authors should include this in their discussion.

3. Page 3: The authors should consider mentioning that clinically relevant “click” chemistry is not limited to azide-alkyne cycloadditions. The first click-based therapy to reach clinical trials is based on inverse-electron demand Diels-Alder click reactions (DOI: 10.1021/ja8053805, review DOI: 10.1016/j.bioorg.2024.107573).

4. Page 3: The authors refer to the polysaccharide coatings as “homogeneous, single-cell coatings.” It would be helpful for the authors to clarify what “homogenous” means for their work.

5. Page 6: Incubation at 4°C is known to reduce internalization and this is a likelier explanation for the reduced intracellular signal from the DBCO/FAM polymers than metabolic activity from the cells.

6. Fig. S12: For ease of viewing, the authors could consider adding labels within the figure (in addition to the caption) to clarify what the rows/columns are for their microscopy images.

7. Fig. S13: Please specify what the red and black dotted lines mean in the figure caption.
8. Page 7: The authors should expand their discussion of relevant literature for how increased transcription of integrins can enhance engraftment, and how long these effects may last.
9. Page 9: The authors suggest that the increase in laminin adhesion is likely due to the "extensive post-translational glycosylation, which confers interactive hydrogen bonding sites for HA compared to collagen". Please clarify what is meant by "extensive post-translational glycosylation". Is this referring to native glycosylation or the exogenous HA/Alg polymers that have been installed on the cell surface?
10. Page 11: In the discussion of the HUVEC adhesion assay, the authors should state that the measured ratios of untreated HPCs to HUVECs in this assay was 42% to put the other values in context.
11. In the ¹H NMR peak assignments of the HA and Alg polymers in the supporting information, the singlet at 2.68 ppm is assigned as an NH peak. This is unlikely as this amide NH should have exchanged with deuterium from the D₂O solvent. This peak also is not present assignments of the FAM-functionalized polymers, which is unlikely if it were in fact the amide NH.
12. In the ¹H NMR peak assignments, DBCO/FAM aromatic peaks are assigned as "C6H6 from DBCO/FAM". These should be corrected to "aromatic CH" for DBCO/FAM.

Minor typos:

1. Page 3, paragraph 2, line 7: please correct "sing-cell" as "single-cell"
2. Page 5, paragraph 1, line 16: please correct "hydrogen binding" to "hydrogen bonding"
3. Page 7, paragraph 2, line 6: please add hyphen to "collagen-coated"
4. Page 9, paragraph 1, line 7: please correct "translate in" to "translate to"
5. Page 22, Acknowledgements, line 9: please correct "GPF" as "GFP"

Reviewer #2

(Remarks to the Author)

In this submission, the authors apply the well-recognized metabolic oligosaccharide engineering (MOE) approach in a novel application, to modify the surface of hepatic progenitor cells (HPCs) with polysaccharides, specifically hyaluronic acid (HA) and alginate (Alg) to determine whether these modifications are likely to improve engraftment of these cells as a cell therapy to address liver disease. If this technique could be used successfully in human clinical applications, it would have significant benefit at addressing a significant need in regenerative medicine, without the need for genetic modification of precursor cells.

On the whole, this is an interesting, generally well-written manuscript with significant novelty. I have the following suggestions/concerns:

1. There are a number of places where the methods descriptions could be more thorough/clear as follows:
 - a. Page 14, indicate vendor/catalog number for DMTMM
 - b. Page 15, indicate vendor/catalog number for FAM
 - c. Page 15, reference supplemental methods for Col-I coated flasks and DMEM medium composition. Also in supplemental methods, indicate volume of collagen solution added to each well of 24-well plates and 35 mm dishes.
 - d. Page 15, clarify what gentle heating means (temperature?) and vortex speed/energy/duration
 - e. Page 15, indicate vendor/catalog number for accutase
 - f. Page 16, what level of confluency was seen when the cells were seeded in the 35-mm dishes? This is significant as it may impact cell health.
 - g. Page 16, indicate catalog number for ECM540 adhesion plates.
 - h. Page 16, for ECM540 assay, were other incubation times beyond 1 hour investigated to see if adhesion was improved? Similarly, why was 2 hours chosen for the HUVEC assay and cell morphology assay. Were longer incubation times evaluated? If incubation times are too short, cells might not be fully attached.
 - i. Page 17, what software was used to quantify morphology by outlining cell perimeters as regions of interest.
 - j. Supplemental information-Page 4, reference supplemental methods for qPCR and statistical analysis in either methods or results (as appropriate) in main document.
2. Figures: Statistical significance should be defined for each set of symbols used on a given figure. E.g., Figure 3 has **, ***, and ****, but only **** P<0.0001 is given in the figure legend.
3. Page 9, the authors write "we analyzed the morphologies of HA and Alg coated cells after 2 h of seeding onto Col-I tissue culture dishes (Fig. 3c, d, and Fig. S18)". Why was Col II not considered since it showed a much larger increase in binding of HA coated cells?
4. Page 11, how (if at all) did the authors validate the interaction identified by flow cytometry as being due to tight cell-cell focal adhesions. Was this confirmed by microscopy?

5. Through figure 2, the authors focus on HA4 and Alg4, then in figure 3, they switch to HA8 and Alg8. Why the change in emphasis. Please clarify in text.
6. In Figure S17, the authors report changes in integrin expression
 - a. Statistical analysis should be performed.
 - b. This is a significant result and the authors seem to gloss over it. While it may be beneficial, clearly a physiological change has resulted from some aspect of the process. Ideally, the authors should provide an experimental explanation, but at the very least, some discussion and potential explanation should be provided in the discussion.
 - c. This change in integrin expression further leads to the question of whether there are any other phenotypic changes in the cells as a result of the surface modulation. While experiments to address that issue may be outside of the scope of this work, this should be included in a discussion of future directions.
6. It would be nice to have biological marker characterization of the liver microtissues in the immune flow chip rather than just DAPI staining.
7. The discussion section is rather superficial. It should address potential limitations of this work, the possible differences between murine cells (used in this study) and human cells, the implications of the change in integrin expression, the impact of the studies, any unexpected results and future directions needed to bring this technology to fruition.

Minor issues:

Page 3, the authors write "homogeneous, sing-cell coating", should be single-cell

Page 12, the authors write "Consistently with these results", should be Consistent with these results

Reviewer #3

(Remarks to the Author)

Overall comment: The study presents a novel, non-genetic method for enhancing hepatic progenitor cell (HPC) engraftment through polysaccharide surface modification. This approach is timely and innovative, addressing the critical challenge of cell adhesion and engraftment that limits the effectiveness of cell-based therapies in regenerative medicine. Although the idea of enhancing cell engraftment through polysaccharide surface modification is attractive in cell therapeutic studies. Some caveats exist in the manuscript as described below. Addressing these points will definitely improve the quality of the manuscript.

1. The manuscript would benefit from clearly separating the 'Introduction' section from the 'Abstract' to enhance readability and clarity. Currently, these sections seem to be combined.
2. In page 11, the population adhesion value (58%) mentioned in the text and the actual data presented in Figure 3b. In Figure 3b, the adhesion value for HPCs functionalized with Alg8 on HUVECs is shown to be 40%, not 58% as stated in the text.
3. It would strengthen the manuscript to include the cell viability assay and integrin gene expression within the main text rather than in the supplementary data. This would allow readers to better evaluate the biocompatibility of the cell modification approach as a central aspect of the study, rather than as additional information.
4. There is limited assessment of the approach's impact on cellular functionality. It would be beneficial to evaluate how this modification affects the functionality of HPC, including their liver-specific functions and differentiation potential towards hepatic lineages. This additional data would provide a more comprehensive understanding of the biocompatibility and efficacy of the approach.
5. It would be worthwhile to discuss the potential of using other biomolecules, such as collagen, chitosan for surface modification. Exploring alternative biomolecules could provide insights into optimizing cell adhesion and engraftment, and may broaden the applicability of this approach.

Version 1:

Reviewer comments:

Reviewer #1

(Remarks to the Author)

The authors did a great job at thoroughly addressing concerns noted and clarifying aspects of the text, figure captions and supplemental methods suggested by the reviewers. I believe the comments from all reviewers have been adequately addressed, and additional data suggested from reviewer 1 and reviewer 3 have been included as well. The inclusion of the controls for non-MOE treated cells (suggested by reviewer 1) is appreciated and emphasizes the significance of MOE to attach the polymers to the cells. I agree that given the minimal background, including these controls for the functional assays would not provide any additional valuable data and therefore am ok that these are not included for Fig 3. Analogously, the functional data provided in Fig 4 d/e addresses the comment from Reviewer 3 asking for data indicating if the author's approach alter cell functionality.

The only suggested correction is for Fig 1a. Many of the structures of the sugars in Fig 1a have been corrected, however the Sia-N-DBCO clicked product (the far right structure, underneath the green box) is the enantiomer of Sia. The orientation should be the same as the SiaNAz to the right of that structure.

I commend and congratulate the authors on their thorough and excellent job at addressing reviewer comments and revising their manuscript.

Reviewer #2

(Remarks to the Author)

In general the authors have responded to the majority of our concerns with one exception, our comment #6 questioning the appropriateness of the hLMTs. While we agree that DAPI staining of the hLMTs is sufficient to demonstrate cell adhesion, a reference to a publication or datasheet which demonstrate the liver tissue mimetic characteristics of hLMTs is needed.

Please include an appropriate reference to this statement in Page 13: "Moreover, the hLMTs are constructed with a range of different cells, including human hepatocytes, Kupfer cells, and lymphatic endothelial cells, mimicking the hepatic microenvironment found in vivo".

In addition, we have the following minor comments on the revised manuscript:

1. Figure S14: Are Alg and HA titles swapped? Because in the Results section, in Page 7 of the manuscript, the authors say "Interestingly, HA-DBCO coated cells exhibited over 100% viability, likely as a consequence of the proliferative effect of HA, previously reported in the literature." However, in the graphs, the Alg groups show >100% viability, not HA.

2. On page 7, authors say "This offers the possibility to shed the polymer coating from the cell membrane in a physiological process, allowing uncoated cells to establish new interactions with the ECM and surrounding tissue in situ." We suggesting call them coating-shed cells or or de-coated cells or something similar instead of uncoated cells (which are another experimental group) to avoid confusion.

3. Figure S18 caption – Cellsens should be changed to cellSens

4. The first paragraph of the discussion has significant redundancy with introduction. We recommend shortening or removing it.

5. Page 15 in Discussion: We suggest changing "Building on this knowledge" to "Building on this hypothesis" in "We hypothesise the attachment of such polysaccharides to the cell surface of HPCs could facilitate non-covalent intermolecular interactions such as electrostatic and hydrogen bonding to carbohydrate and protein moieties on target endothelial surfaces. Building on this knowledge, we envisioned the use of polysaccharides as a proof of concept for the development of bioadhesive coatings for cell-based therapies."

Also hypothesize is misspelled here.

6. On Page 17 in the Discussion section, in the last line "evidenced by recent applications as diagnostics, prodrugs", the word "such" is missing before "as diagnostics, prodrugs"

Reviewer #4

(Remarks to the Author)

We thank all reviewers for their constructive feedback. We have done our best to fully address their comments and we hope that the manuscript has improved as a result.

Reviewer #1:

1. Controls for HA- or Alg-polymer treated HPCs alone (i.e. in the absence of MOE) are not provided, with the exception of data shown in Figure S12c/d for the 4°C SPAAC. On page 6 the authors state that at 37°C there is “extensive internalization of the biopolymers and non-specific membrane interactions even in in non-azide treated cells”, but this control data is not provided.

These data have now been added in Fig. S12 and S13.

2. Further, it does appear that some fluorescein signal in the HA8- and Alg8-treated polymer control samples at 4°C SPAAC is present, particularly for the Alg-polymer control (Figure S12c/d), suggesting there may be background or non-specific adherence of the polymers under these conditions. These controls are important to include in flow cytometry data in Fig. 1c-f to better quantify the background signal of the polymer.

The non-specific fluorescence observed in Fig. S13c, S13d (previously S12c, S12d) in non-azide treated cells is minimal in comparison to the fluorescence detected at 4 °C in azide-treated samples. Moreover, fluorescence does not appear to be localised at the cell membrane, as shown in the brightfield images. This suggests that a small amount of polymer is aggregating in solution rather than binding to the cell membrane. Additional flow cytometry data have been added in Fig. 1c, 1d to complement confocal fluorescence images.

3. If significant background non-specific binding of the DBCO-polymers is observed, these polymer-alone controls (i.e. non-MOE treated HPCs) would be needed in the functional assays, such as the HUVEC adhesion assay in Fig. 3b, to demonstrate that the covalent attachment of the polymers to SiaNAz-engineered cells is required for the adhesion they are reporting.

As clarified in point 2, the non-specific binding we observed is minimal and does not appear to be localised at the cell surface (Fig. S13). Moreover, even if some non-MOE treated cells were to be coated, polymers would only be weakly bound to the surface through non-covalent interactions (rather than covalent bonds as allowed by MOE), meaning they would be more susceptible to removal upon introduction to dilute conditions and exposure to vascular shear forces *in vivo*. Based on this reasoning, we believe that the addition of functional assays to test these experimental groups would provide limited value for the intended application of our coated cells.

4. The discussion and interpretation data of the polymer lifetime-display data (page 7 discussion; Data Fig. 2 and Fig S16) could be expanded and made more clear for how the authors are coming to their conclusions stated. It is suggested that the coatings “started to visibly detach from the surface after 24 h.” What evidence suggests that the coatings are visibly detaching? Is it known whether the polymer is being shed, or if the glycoconjugate anchors are being internalized and digested/recycled within the cell and are not present on the cell membrane?

We agree with this reviewer that the language we used in this instance is confusing. We modified the text as follows: “The fluorescence intensity at the cell surface started to decrease after 24 h of incubation at 37 °C, 5% CO₂, when cells were left to adhere on a collagen-coated dish. Complete

disappearance of the fluorescence signal could be observed after 96 h, as evidenced by confocal fluorescence microscopy (Fig. 2a, b, and Fig. S17-S18) and flow cytometry (Fig. 2c), with a level of quantified fluorescence comparable to uncoated cells at the end of this time window. As expected, no difference was observed between the two polysaccharide coatings or among the different degrees of DBCO functionalization. Internalization of the biopolymers was observed by analyzing intracellular slices of confocal Z-stack images, with intracellular fluorescence decreasing significantly after 24 h (Fig. S19-S20). Based on these data and previous reports, it is likely that the glycoconjugate anchors are being internalized and processed within the cell at a rate that is solely dependent on the specific membrane turnover of each cell population (*Biomacromolecules* 2019, **20**, 2726-2736; *Acs Macro Lett* 2018, **7**, 1289-1294; *Science* 2008, **320**, 664-667; *J Inherit Metab Dis* 2018, **41**, 515-523)."

5. After 6 h, it appears that some internalization can be seen by confocal microscopy. At 24 h and beyond, the FAM signal appears quite punctate and it is unclear whether the signal is restricted to the cell surface or is intracellular. Analysis of the Z-stacks of the confocal microscopy data should help clarify whether the signal is intracellular or restricted to the surface.

We thank the reviewer for this suggestion. We analysed the z-stacks from Figs. 2a and 2b and S17 to determine the localisation of the fluorescent signals. At 6 h post coating, fluorescence was predominantly still localised at the cell surface. However, Z-slices aligned with nuclear centre revealed that some cells had already started to show internalisation (Fig. S19 and S20).

By 24 h, cells fully adhered and flattened on the Col-I surface, with the fluorescence signal becoming punctate and showing the polymer is internalized. This has now been clarified in the text and the relevant figures have been added to the SI (Fig. S19 and S20).

6. Additionally, the Fig. 2d caption for the flow cytometry data specifies it is cell-surface fluorescence being observed; however, if the polymers are internalized (e.g. present in lysosomes), but not degraded, would a fluorescence signal not still be observed for the internalized polymer? It would be helpful for the authors to clarify these points to support their conclusions.

We thank the reviewer for this comment. Fig. 2 caption has now been revised to state "cell fluorescence" to accurately reflect the measurement of total cell fluorescence (not only cell-surface) by flow cytometry.

7. In the discussion section (page 14), the authors state that "the transient nature of the conjugated polysaccharide functionalities aligns well with the regenerative timescale." What is this timescale? It would be useful for the authors to comment on what the desired lifetime of display of their polymer would be for their engraftment-enhancing functions and for future translation into a regenerative therapy (i.e. if their polymer is still present on the surface, albeit at a reduced amount due to turnover and internalization, is this timeline translatable?).

We agree with the reviewer that this claim needs clarification, which has now been added in the discussion section. This statement was based on prior studies in literature where hepatic progenitor cells (and hepatocyte-based cell replacement therapies) were shown to engraft within 1-4 days post-transplantation *in vivo* (*Sci. Rep.* 2016, **6**, 21783; *Liver Transpl.* 2015, **21**, 652-661; *Cell Transplant.* 2021, **30**:963689721993780; *Cell Stem Cell.* 2022, **29**, 355-371). This suggests that the critical window for polymer-assisted engraftment falls within this timescale.

Our polymer is designed to remain present on the cell surface long enough to support initial interaction and adhesion to the endothelium while allowing for natural turnover and internalization of the membrane. The transient nature of the conjugated polysaccharide aligns with the short engraftment period, facilitating initial attachment without long-term interference in other functions and processes, including differentiation as demonstrated in Fig. 4d and 4e.

8. The authors characterize their functionalized polysaccharides using a combination of ^1H NMR spectrometry, UV-vis spectroscopy, and size exclusion chromatography. For the HA polymers, integration of ^1H NMR signals for DBCO relative to the acetamide singlet gives a reasonable estimate for DBCO functionalization, as has been done in similar literature (DOI: 10.1007/s10856-023-06757-9). However, without a clear acetamide peak for Alg, the overlapping signals from 3.7-4.2 ppm were for relative integration (integrating to 4 as the H2-H5 of the monosaccharide eating unit); because of the broadening of signals due to the polymeric nature of Alg, this may not accurately quantify the DBCO functionalization. It would strengthen the authors claims to consider the following:

a. Does integration of H2-H6 region of the DBCO-HA (~3.2-4.0 ppm) align with the 4% and 8% DBCO functionalization quantified by the acetamide singlet integration (2 ppm)? If yes, this suggests the quantification of the DBCO-Alg polymers is accurate.

Following integration of the H2-H6 region (see HA4-DBCO ^1H NMR spectrum below for reference), we confirm that the quantification of the DBCO-HA polymer is accurate. This value of ~10 protons remains consistent when integrating the same region in HA8-DBCO. Fig. S4 and S5 have been updated to include this integration.

Additionally, the signal in the aromatic region is very low in Figs. S5/7; more scans should be acquired to increase signal beyond background noise to confirm DBCO/FAM quantification.

In fully functionalised HA and Alg polymers ^1H NMR spectroscopy signals for DBCO and FAM overlap in the aromatic region. Acquiring spectra with longer scans (or higher number of scan) has therefore not helped with FAM quantification. For this reason, we quantified FAM attachment through UV-Vis absorption rather than NMR spectroscopy.

b. DBCO absorbs at 290 and 310 nm? Can the authors use this absorbance to quantify DBCO-HA/Alg functionalization by UV-vis spectroscopy, similar to FAM quantification?

Yes, it is correct that DBCO absorbs in the above range. However, we are already able to accurately quantify DBCO conjugation using ^1H NMR spectroscopy (see response to point a).

9. The methods are well written and fully describe the experimental protocols for the data provided. The manuscript would be strengthened by clarifying and including more details of the experiment conditions in the figure captions and in the figures. For example, given the differences in display of DBCO-HA/Alg depending on time and temperature of incubation, it should be clear in the figure and figure caption what time and temperature were used (Fig. 1b, S12). Additionally, the number of licates, error bars, and details statistical analyses were not provided for some experiments (e.g. Fig. 3b, 4c). Were these experiments replicated? The authors should expand figure captions and labels within figures to ensure that all relevant experimental details and data are included.

The details requested (time points and temperature as well as details of statistical analysis) have been added to the figure captions. Moreover, all the figure captions have been thoroughly checked and populated with further details as suggested.

10. The sialic acid structure in Fig. 1a should be corrected. The authors should ensure the correct stereochemistry is displayed at C2 and C5, and the “Ac” on the exocyclic N should be “H”. In MOE literature, the chemical structures of sugars like ManNAz and sialosides are commonly drawn in a chair conformation (see Fig. 2C in DOI: 10.1042/BCJ20200612 for an example).

We thank the reviewer for pointing this out. Sialic acid structures have now been corrected to show the correct chair conformations.

11. The Fig. 1 caption should be corrected to reflect that NeuAz is incorporated on glycan structures in lieu of sialic acid residues, not on Sia-containing residues.

Fig. 1 caption has been corrected as suggested.

12. The authors could consider providing additional details in the introduction to expand on the rationale for remodelling cell surfaces with Alg and HA for engraftment. It is noted that Alg and HA can interact with ECM proteins and free glycans, but more mechanistic context for their role in maintaining/regulating cell adhesion would be helpful. It would also be useful to include the general repeating structures of HA/Alg in Figure 1.

A sentence has been added in the introduction to better explain the role of Alg and HA in maintaining cell adhesion. The paragraph now reads as follows: “In particular, we explore

hyaluronic acid (HA) and alginate (Alg) as exogenous cell surface functionalities, owing to their ability to establish hydrogen bonds with ECM proteins and glycans. These biopolymers have been previously exploited for bioadhesive and targeting applications in drug and cell-delivery (*Hepatology* 2013, 57, 775-784; *Sci. Pharm.* 2010, 78, 941-957; *J. Mech. Behav. Biomed. Mater.* 2023, 143; *Stem cell res. ther.* 2019, 10; *Matter* 2024, 7, 3447-3468). We hypothesize that coating the cell membrane with these polysaccharides could enhance adhesion by facilitating hydrogen bonding at the endothelial interface, mimicking the rolling mechanisms observed in native carbohydrate-mediated interactions.”

Structures for the repeating units of HA and Alg have been added to Fig. 1.

13. Page 2, Ref 12 suggests low engraftment is due to more than inadequate adhesion to the sinusoidal epithelium (oxidative stress, immune activation, etc). For accuracy, the authors should include this in their discussion.

This has now been added to the relevant section, as suggested.

14. Page 3: The authors should consider mentioning that clinically relevant “click” chemistry is not limited to azide-alkyne cycloadditions. The first click-based therapy to reach clinical trials is based on inverse-electron demand Diels-Alder click reactions (DOI: 10.1021/ja8053805, review DOI: 10.1016/j.bioorg.2024.107573).

We thank the reviewer for their suggestion. This has now been added to our revised discussion section.

15. Page 3: The authors refer to the polysaccharide coatings as “homogeneous, single-cell coatings.” It would be helpful for the authors to clarify what “homogenous” means for their work.

We have now clarified in the text what “homogeneous coating” means in this context.

16. Page 6: Incubation at 4°C is known to reduce internalization and this is a likelier explanation for the reduced intracellular signal from the DBCO/FAM polymers than metabolic activity from the cells.

This is indeed why we used low-temperature incubation. This point has been clarified in the text, and a relevant reference has been added.

17. Fig. S12: For ease of viewing, the authors could consider adding labels within the figure (in addition to the caption) to clarify what the rows/columns are for their microscopy images.

Relevant labels in Fig. S12 and S13 have now been added.

18. Fig. S13: Please specify what the red and black dotted lines mean in the figure caption.

The meaning of the red and black dotted lines in Fig. S14 (previously S13) has been clarified in the figure caption.

19. Page 7: The authors should expand their discussion of relevant literature for how increased transcription of integrins can enhance engraftment, and how long these effects may last.

A discussion around the link between transcription of integrins and cell engraftment has been added, as suggested, in the ‘Results’ and in the revised ‘Discussion’ sections.

20. Page 9: The authors suggest that the increase in laminin adhesion is likely due to the “extensive post-translational glycosylation, which confers interactive hydrogen bonding sites for HA compared to collagen”. Please clarify what is meant by “extensive post-translational glycosylation”. Is this referring to native glycosylation or the exogenous HA/Alg polymers that have been installed on the cell surface?

This specifically refers to the native post translational glycosylation of laminin, tenascin and vitronectin, which provides potential sites of dynamic hydrogen bond interactions with the polysaccharide coatings. This has now been clarified in the text.

21. Page 11: In the discussion of the HUVEC adhesion assay, the authors should state that the measured ratios of untreated HPCs to HUVECs in this assay was 42% to put the other values in context.

As suggested, this has now been added to the text in the relevant section.

22. In the ¹H NMR peak assignments of the HA and Alg polymers in the supporting information, the singlet at 2.68 ppm is assigned as an NH peak. This is unlikely as this amide NH should have exchanged with deuterium from the D₂O solvent. This peak also is not present assignments of the FAM-functionalized polymers, which is unlikely if it were in fact the amide NH.

We thank the reviewer for spotting this mistake. The peak at 2.68 is attributed to residual DMSO from the DBCO functionalisation, as correctly labelled in Fig. S1-S8. This has now been corrected.

23. In the ¹H NMR peak assignments, DBCO/FAM aromatic peaks are assigned as “C₆H₆ from DBCO/FAM”. These should be corrected to “aromatic CH” for DBCO/FAM.

As suggested, this has now been corrected.

Minor typos:

1. Page 3, paragraph 2, line 7: please correct “sing-cell” as “single-cell”
2. Page 5, paragraph 1, line 16: please correct “hydrogen binding” to “hydrogen bonding”
3. Page 7, paragraph 2, line 6: please add hyphen to “collagen-coated”
4. Page 9, paragraph 1, line 7: please correct “translate in” to “translate to”
5. Page 22, Acknowledgements, line 9: please correct “GPF” as “GFP”

We thank this reviewer for highlighting these typos, which have all been corrected.

Reviewer #2:

1. There are a number of places where the methods descriptions could be more thorough/clear as follows:

a. Page 14, indicate vendor/catalog number for DMTMM

This has now been added in the relevant section.

b. Page 15, indicate vendor/catalog number for FAM.

This has now been added in the relevant section.

c. Page 15, reference supplemental methods for Col-I coated flasks and DMEM medium composition. Also in supplemental methods, indicate volume of collagen solution added to each well of 24-well plates and 35 mm dishes.

These details have now been added in the relevant sections.

d. Page 15, clarify what gentle heating means (temperature?) and vortex speed/energy/duration

This has now been clarified in the relevant section.

e. Page 15, indicate vendor/catalog number for accutase

This has now been added in the relevant section.

f. Page 16, what level of confluency was seen when the cells were seeded in the 35-mm dishes? This is significant as it may impact cell health.

Confluency upon seeding was 20%. The HPCs used in this study can tolerate much lower seeding densities (up to 250 cells \times cm²) due to their high clonogenic potential, as demonstrated by Lu *et al.* (*Protoc. Exch.* 2015). This has now been added in the relevant section.

g. Page 16, indicate catalog number for ECM540 adhesion plates.

This has now been added in the relevant section.

h. Page 16, for ECM540 assay, were other incubation times beyond 1 hour investigated to see if adhesion was improved? Similarly, why was 2 hours chosen for the HUVEC assay and cell morphology assay. Were longer incubation times evaluated? If incubation times are too short, cells might not be fully attached.

We wanted to investigate the initial interactions between coated cells and ECM proteins/glycoproteins. This is particularly helpful when moving from an *in vitro* to an *in vivo* setting, with the hypothesis that our coatings are able to increase cell-ECM interactions post-injection, retaining as many cells as possible at the desired site. For this reason, a relatively short incubation time was selected. This is also in line with the ECM540 kit manufacturer protocol, which recommends a 1 h incubation to study cell-ECM protein interactions. Indeed, even shorter incubation times of up to 30 min have previously been used for similar adhesion assays (*Biotechniques* 2020, 68, 325-333).

With regards to the HUVEC assays, we found that a longer 2 h incubation period was the minimum time necessary for HPCs to display attachment and subsequent changes in cell morphology. This endpoint aligns with the biological timescale required for signal transduction and cytoskeletal rearrangement. We found that incubation times of 1 h resulted in less adhesion and reduced morphological change as a result of a less robust binding, as stated by the reviewer.

The main manuscript has been modified to include justification for the time points selected.

i. Page 17, what software was used to quantify morphology by outlining cell perimeters as regions of interest.

The software used was FIJI. This information has been added to the methods section.

j. Supplemental information-Page 4, reference supplemental methods for qPCR and statistical analysis in either methods or results (as appropriate) in main document.

Reference to qPCR methods and statistical analysis has now been added to the main manuscript (results section).

2. Figures: Statistical significance should be defined for each set of symbols used on a given figure. E.g., Figure 3 has **, ***, and ****, but only **** P<0.0001 is given in the figure legend.

Statistical significance has been defined for each symbol.

3. Page 9, the authors write "we analyzed the morphologies of HA and Alg coated cells after 2 h of seeding onto Col-I tissue culture dishes (Fig. 3c, d, and Fig. S18)". Why was Col II not considered since it showed a much larger increase in binding of HA coated cells?

Col-I was selected because it is the predominant collagen isotype in the liver ECM, playing a critical structural and functional role. Its abundance also increases significantly during the progression of liver fibrosis, making it a representative substrate for studying cell adhesion to fibrotic tissue (*PLOS ONE* 2016, 11, 3, e0151736). While Col-II showed a larger increase in the binding of HA-coated cells in our ECM adhesion assay, its expression in liver ECM is minimal compared to Col-I. Thus, using Col-II in our assays may not accurately reflect the native liver environment. This clarification has now been added in the main manuscript.

4. Page 11, how (if at all) did the authors validate the interaction identified by flow cytometry as being due to tight cell-cell focal adhesions. Was this confirmed by microscopy?

We complemented the flow cytometry analysis with confocal microscopy to validate the interactions identified as being due to the formation of robust cell-cell interaction and adhesion to the monolayer. Using GFP-labelled HPCs, we visualised their interactions with HUVEC monolayers and observed distinct morphological features indicative of focal adhesion formation. These features included flattening and spreading of HPCs on the HUVEC monolayer. To further quantify these interactions, we analysed morphological parameters such as cell circularity and area. These findings, detailed in Fig. 3e, 3f and Fig. S22, corroborate the flow cytometry data and confirm the presence of strong cell-cell adhesions after extensive wash steps. We have removed the word "tight" to remove potential confusion with tight junctions.

5. Through figure 2, the authors focus on HA4 and Alg4, then in figure 3, they switch to HA8 and Alg8. Why the change in emphasis. Please clarify in text.

In terms of adhesive properties, HA4 and Alg4 coatings behave similarly to Alg8 and HA8 coatings, and therefore data have been presented interchangeably throughout the paper. However, we recognise this can be confusing and have now replaced the Alg4/HA4 data in Fig. 2 with those for Alg8 and HA8 coatings. We avoided to modify Fig. 1c and 1d, where showing flow cytometry data of Alg4 and HA4 coatings helps us to demonstrate the effectiveness of the click functionalities even at lower DBCO functionalisation. Additional flow cytometry data for Alg8 and HA8 coatings are shown in Fig. S15.

6. In Figure S17, the authors report changes in integrin expression

a. Statistical analysis should be performed.

Integrin expression profiles have now been added to Fig.2 with statistical analysis.

b. This is a significant result and the authors seem to gloss over it. While it may be beneficial, clearly a physiological change has resulted from some aspect of the process. Ideally, the authors should provide an experimental explanation, but at the very least, some discussion and potential explanation should be provided in the discussion.

We agree with the reviewer on this point. A further discussion on the integrin expression results has been added to the main text.

c. This change in integrin expression further leads to the question of whether there are any other phenotypic changes in the cells as a result of the surface modulation. While experiments to address that issue may be outside of the scope of this work, this should be included in a discussion of future directions.

We appreciate the reviewer's insightful comment. In this study, we focused on integrin expression as it directly relates to our primary interest in understanding and enhancing cell adhesion. While we did not assess any other potential phenotypic changes induced by the surface coating, we acknowledge that such changes may occur and could have implications for cell behaviour and function. We have added a sentence in the discussion section of the manuscript highlighting the possibility of other phenotypic changes and suggesting that this is an important area for future investigation.

6. It would be nice to have biological marker characterization of the liver microtissues in the immune flow chip rather than just DAPI staining.

The hLMTs used in this study were selected as representative binding partners of hepatic cell types that injected HPCs are likely to encounter *in vivo*. Our primary goal was to assess functional outcomes (adhesion) rather than perform a comprehensive characterization of the microtissue environment. The species difference between the HPCs (mouse-derived) and the hLMTs (human-derived) may also limit the protein-protein interactions between these two models (*e.g.* vinculin formation at the cell boundaries). This limitation reflects the early stage of our approach.

Nonetheless, the adhesion observed is still biologically relevant, as also demonstrated in a recent study where similar human-derived biliary HPCs could engraft within mouse models of biliary disease, highlighting the potential for cross-species compatibility in such interactions (*Cell Stem Cell* 2022, 29, 355-371). In future studies, we aim to use human-human systems and incorporate additional marker analyses, such as immunostaining for liver-specific proteins, as well as monitoring of focal adhesion formation to better elucidate the molecular mechanisms underlying adhesion processes.

7. The discussion section is rather superficial. It should address potential limitations of this work, the possible differences between murine cells (used in this study) and human cells, the implications of the change in integrin expression, the impact of the studies, any unexpected results and future directions needed to bring this technology to fruition.

We thank the reviewer for this comment. We have significantly expanded the discussion section to embed all the suggested topics.

Minor issues:

Page 3, the authors write "homogeneous, sing-cell coating", should be single-cell.

Page 12, the authors write "Consistently with these results", should be Consistent with these results
We thank the reviewer for highlighting these typos which have now been corrected.

Reviewer #3:

1. The manuscript would benefit from clearly separating the 'Introduction' section from the 'Abstract' to enhance readability and clarity. Currently, these sections seem to be combined.

The abstract and introduction sections have now been labelled to allow a clear distinction between them.

2. In page 11, the population adhesion value (58%) mentioned in the text and the actual data presented in Figure 3b. In Figure 3b, the adhesion value for HPCs functionalized with Alg8 on HUVECs is shown to be 40%, not 58% as stated in the text.

We thank the reviewer for spotting this oversight. This has now been corrected in the text.

3. It would strengthen the manuscript to include the cell viability assay and integrin gene expression within the main text rather than in the supplementary data. This would allow readers to better evaluate the biocompatibility of the cell modification approach as a central aspect of the study, rather than as additional information.

The integrin expression data have now been moved into the main text (Fig. 2d). With regards to the cell viability data, we strongly feel they find better place in the supplementary information. Indeed, the cytocompatibility of alginate and hyaluronic acid has been extensively assessed with a wide range of cell types and hence these data do not report any new findings compared to existing literature.

4. There is limited assessment of the approach's impact on cellular functionality. It would be beneficial to evaluate how this modification affects the functionality of HPC, including their liver-specific functions and differentiation potential towards hepatic lineages. This additional data would provide a more comprehensive understanding of the biocompatibility and efficacy of the approach.

We thank the reviewer for this suggestion. We have now performed an additional experiment to ascertain the retained differentiation potential of HPCs. Cells were coated and left for 4 days to allow for the membrane turnover and disappearance of fluorescence as previously observed in our coating lifetime assessments. *In vitro* differentiation was then performed by the addition of Wnt3a and 1% DMSO to the media at day 5. Cell morphology was monitored daily for 6 days and media changed at 3 days. Medium was kept for albumin quantification by ELISA. After treatment, cells were then shown to increase in size and became hexagonal in shape, with increased glycogen storage by PAS staining. Murine albumin in the culture medium was also shown to increase during treatment. Additional text has now been added in the "Results" and "Methods" sections to reflect this change.

5. It would be worthwhile to discuss the potential of using other biomolecules, such as collagen, chitosan for surface modification. Exploring alternative biomolecules could provide insights into optimizing cell adhesion and engraftment, and may broaden the applicability of this approach. We thank the reviewer for this thoughtful suggestion. The MOE approach described in the manuscript is indeed versatile and can be adapted to a wide range of biomolecules for surface modification. We have added a paragraph in the discussion section to emphasise the potential of exploring alternative biomolecules and other cell-therapies.

Reviewer #1:

The authors did a great job at thoroughly addressing concerns noted and clarifying aspects of the text, figure captions and supplemental methods suggested by the reviewers. I believe the comments from all reviewers have been adequately addressed, and additional data suggested from reviewer 1 and reviewer 3 have been included as well. The inclusion of the controls for non-MOE treated cells (suggested by reviewer 1) is appreciated and emphasizes the significance of MOE to attach the polymers to the cells. I agree that given the minimal background, including these controls for the functional assays would not provide any additional valuable data and therefore am ok that these are not included for Fig 3. Analogously, the functional data provided in Fig 4 d/e addresses the comment from Reviewer 3 asking for data indicating if the author's approach alter cell functionality.

I commend and congratulate the authors on their thorough and excellent job at addressing reviewer comments and revising their manuscript.

We thank the reviewer for their constructive feedback and for recognising our efforts in addressing the reviewer's concerns.

The only suggested correction is for Fig 1a. Many of the structures of the sugars in Fig 1a have been corrected, however the Sia-N-DBCO clicked product (the far right structure, underneath the green box) is the enantiomer of Sia. The orientation should be the same as the SiaNAz to the right of that structure.

We thank the reviewer for pointing this out. We have amended the stereochemistry of the Sia-N-DBCO clicked product to reflect the same enantiomer in SiaNAz.

Reviewer #2:

In general the authors have responded to the majority of our concerns with one exception, our comment #6 questioning the appropriateness of the hLMTs. While we agree that DAPI staining of the hLMTs is sufficient to demonstrate cell adhesion, a reference to a publication or datasheet which demonstrate the liver tissue mimetic characteristics of hLMTs is needed. Please include an appropriate reference to this statement in Page 13: "Moreover, the hLMTs are constructed with a range of different cells, including human hepatocytes, Kupfer cells, and lymphatic endothelial cells, mimicking the hepatic microenvironment found in vivo".

We have added a reference to this paragraph, as suggested by this reviewer.

In addition, we have the following minor comments on the revised manuscript:

1. Figure S14: Are Alg and HA titles swapped? Because in the Results section, in Page 7 of the manuscript, the authors say "Interestingly, HA-DBCO coated cells exhibited over 100% viability, likely as a consequence of the proliferative effect of HA, previously reported in the literature." However, in the graphs, the Alg groups show >100% viability, not HA.

We thank the reviewer for spotting this oversight. The figure caption has now been updated to reflect the correct set of data.

2. On page 7, authors say “This offers the possibility to shed the polymer coating from the cell membrane in a physiological process, allowing uncoated cells to establish new interactions with the ECM and surrounding tissue in situ.” We suggesting call them coating-shed cells or or de-coated cells or something similar instead of uncoated cells (which are another experimental group) to avoid confusion.

We have modified “uncoated cells” in this paragraph with “de-coated cells”.

3. Figure S18 caption - Cellsens should be changed to cellSens

This has now been modified in Figure S18 caption.

4. The first paragraph of the discussion has significant redundancy with introduction. We recommend shortening or removing it.

The first paragraph of the discussion has been shortened to: “In this study, we used metabolic oligosaccharide engineering and bio-orthogonal click chemistry to functionalize regenerative hepatic progenitor cells with polysaccharides, with the aim to control cell adhesion mechanisms through targeted cell membrane modification.”

5. Page 15 in Discussion: We suggest changing “Building on this knowledge” to “Building on this hypothesis” in “We hypothesise the attachment of such polysaccharides to the cell surface of HPCs could facilitate non-covalent intermolecular interactions such as electrostatic and hydrogen bonding to carbohydrate and protein moieties on target endothelial surfaces. Building on this knowledge, we envisioned the use of polysaccharides as a proof of concept for the development of bioadhesive coatings for cell-based therapies.” Also hypothesize is misspelled here.

We have replaced “knowledge” with “hypothesis” in this paragraph as indicated by this reviewer. We have also corrected the spelling of “hypothesize”.

6. On Page 17 in the Discussion section, in the last line “evidenced by recent applications as diagnostics, prodrugs”, the word “such” is missing before “as diagnostics, prodrugs”.

We have added “such” before “as” in this sentence.